# Lipid droplet and peroxisome biogenesis occur at the same ER subdomains

Amit S. Joshi[1], Benjamin Nebenfuehr[1], Vineet Choudhary[1], Prasanna Satpute-Krishnan[2], Tim P. Levine [3], Andy Golden[1] & William A. Prinz[1]

Nascent lipid droplet (LD) formation occurs in the endoplasmic reticulum (ER) membrane but it is not known how sites of biogenesis are determined. We previously identified ER domains in *S. cerevisiae* containing the reticulon homology domain (RHD) protein Pex30 that are regions where preperoxisomal vesicles (PPVs) form. Here, we show that Pex30 domains are also sites where most nascent LDs form. Mature LDs usually remain associated with Pex30 subdomains, and the same Pex30 subdomain can simultaneously associate with a LD and a PPV or peroxisome. We find that in higher eukaryotes multiple C2 domain containing transmembrane protein (MCTP2) is similar to Pex30: it contains an RHD and resides in ER domains where most nascent LD biogenesis occurs and that often associate with peroxisomes. Together, these findings indicate that most LDs and PPVs form and remain associated with conserved ER subdomains, and suggest a link between LD and peroxisome biogenesis.

[1] National Institute of Diabetes and Digestive and Kidney Diseases, NIH, Bethesda, MD 20892, USA. [2] Uniformed Services, University of the Health Sciences, Bethesda, MD 20814, USA. [3] University College London, Institute of Ophthalmology, London EC1V 9EL, UK. Correspondence and requests for materials should be addressed to A.S.J. (email: amit.joshi@nih.gov) or to W.A.P. (email: prinzw@helix.nih.gov)

L ipid droplets (LDs) play critical roles in cellular metabolism. They are storage depots containing neutral lipids, primarily triacylglycerols (TAGs), and steryl esters (SEs), which can be used for energy production and lipid metabolism. The neutral lipid core of LDs is covered by a phospholipid monolayer that contains coat proteins like perilipins and lipid metabolism enzymes[1–6]. The formation of nascent LDs occurs in the endoplasmic reticulum (ER), but the mechanism remains poorly understood. Nascent droplets initially form between the leaflets of the ER membrane and bud from the membrane as they grow[6,7]. How sites of LD formation are determined and whether these sites form stocastically or at stable, specialized regions of the ER is not known, though there is some evidence that LDs form at preexisting sites[8–10]. It is also possible that nascent LDs initially form stochastically throughout the ER, and then diffuse to specialized sites where they mature and grow. A few proteins are thought to play direct roles in LD biogenesis, though their functions are not well understood. One of these proteins is seipin, called Sei1 in S. cerevisiae, which is expressed in all cell types and localizes to ER-LD contacts where it may facilitate membranous ER-LD connections[10–13].

Peroxisome biogenesis also occurs in the ER. Peroxisomes are single membrane-bound organelles that have important roles in the metabolism of lipids, polyamines, D-amino acids, and fatty acid β oxidation[14]. Vesicles termed pre-peroxisome vesicles (PPVs) bud from the ER and mature into functional peroxisomes. There has been disagreement about whether more than one type of PPV is generated in the ER. There is also evidence that in mammalian cells some PPV biogenesis occurs in the mitochondrial outer membrane[15].

Pex3 and Pex19 are two proteins that play a role in targeting membrane-embedded proteins to PPVs or peroxisomes. Although it had been thought that cells lacking either of these proteins were devoid of peroxisomes and PPVs, a few years ago the van der Klei group showed that yeast cells lacking both Pex3 and Atg1, which is necessary for autophagy, contain a small number of PPVs[16]. These PPVs are normally degraded by autophagy. PPV biogenesis can therefore be studied in cells lacking Pex3 and Atg1 (pex3Δ atg1Δ). We have previously shown that these PPVs originate in the ER, probably at ER domains containing the protein Pex30[17].

Pex30 has an N-terminal reticulon homology domain (RHD) [17]. Reticulons and reticulon-like proteins are abundant conserved ER-shaping membrane proteins that stabilize the highly curved portions of the ER, tubules, and the edges of ER sheets through the RHDs forming wedge-shaped hydrophobic hairpins[18]. We found that overexpression of the RHD domain of Pex30 restores ER structure in S. cerevisiae cells lacking the reticulons. Endogenously expressed Pex30 is in ER subdomains in tubules and the edges of sheets, as are reticulons[17]. The function of Pex30 is not known but it may play a role in peroxisome biogenesis since the size and number of peroxisomes is altered in cells lacking Pex30[19]. Pex30 has also been suggested to reside at ER-peroxisome contacts[20,21].

We wondered whether LD biogenesis occurs at Pex30 domains in the ER for two reasons. First, there are about tenfold more Pex30 domains than there are PPVs in cells[17], suggesting these domains have other functions. Second, recent evidence suggests that some proteins play dual roles in the biogenesis of both LDs and peroxisomes; the Kopito group found that Pex3 and Pex19 insert membrane-embedded proteins into the surface of LDs at ER subdomains[22]. Here, we show that most de novo LD biogenesis occurs at Pex30 subdomains. We also demonstrate that in higher eukaryotes, the protein multiple C2 domain containing transmembrane protein (MCTP2) has a RHD similar to that of Pex30. Interestingly,

peroxisomes and LDs were frequently found to associate with Pex30/MCTP2 domains.

## Results

**Most nascent LDs are associated with Pex30 subdomains.** To determine whether nascent LDs mature at Pex30 subdomains, we visualized LD biogenesis in a S. cerevisiae strain in which LD formation can be controlled. Four enzymes produce neutral lipids in this yeast: Are1 and Are2, which generate SE, and Lro1 and Dga1, which synthesize TAG. Cells lacking all four proteins lack neutral lipids and LDs[23]. We used a strain in which the galactose regulatable promoter GAL1 controls expression of LRO1 and the other three neutral lipid-synthesizing enzymes are not produced (GAL1-LRO1 3Δ). When this strain is grown in a medium containing raffinose, it lacks LDs but begins to produce them when galactose is added[23]. The strain also expressed Pex30-2xmCherry and Erg6-GFP, a LD marker. Before LRO1 induction, Erg6-GFP is on the ER, but it localizes to LDs after galactose addition (Fig. 1a; Supplementary Fig. 1c). About 70% of Erg6-GFP punctae colocalize or are closely associated with Pex30-2xmCherry (Fig. 1b). Similar results were obtained when nascent LDs were visualized with the lipophilic dye BODIPY (Supplementary Fig. 1A).

To verify that most nascent LDs mature at Pex30 subdomains, we induced LD production with a second method. When oleic acid is added to growing cells, they begin to produce new LDs within 30–60 min. We added oleic acid to cells expressing Dga1-GFP from a high copy plasmid and endogenously expressed Pex30-2xmCherry. In growing cells, Dga1-GFP is in the ER, but it relocates to the surface of LDs when LD production is induced[23,24]. We found that all Dga1-GFP puncta colocalize with Pex30-2xmCherry within 1–2 h after oleic addition (Fig. 1c). Lro1-GFP similarly accumulates at sites containing Pex30-2xmCherry after oleic acid addition. (Supplementary Fig. 1B). Together, these findings indicate that most nascent LDs colocalize with Pex30 subdomains in the ER.

**LD induction alters lipid composition at Pex30 subdomains.** If Pex30 subdomains are sites of LD biogenesis, we speculated that they might become enriched in neutral lipids or their precursors, such as diacylglycerol (DAG), when LD production is induced. To investigate this possibility, we used an ER-DAG sensor to measure the distribution of DAG in the ER that we have described and characterized in a previous study[25]. The ER-DAG sensor has the DAG-binding tandem C1 domains of protein kinase D (C1a/b-PKD) fused to GFP and the transmembrane domain of Ubc6, a tail-anchored ER protein. The sensor localizes to the ER and becomes enriched in puncta in the ER that colocalize with Yft2, an ER protein that becomes enriched at LD biogenesis sites when LD production is induced[25]. Previous studies have shown that C1a/b-PKD binds specifically to DAG in vitro and in cells[26–28]. In addition, we found that ER-DAG sensor enrichment at ER puncta during LD biogenesis is blocked by a point mutation in C1a/b-PKD known to ablate DAG-binding in vitro, suggesting that membrane-binding by the sensor in cells is responsive to DAG levels[25,26].

When the cells were grown in regular media, the sensor was uniformly present in the ER, but it became highly enriched in portions of the ER within 1–2 h after oleic acid addition, forming bright puncta. About 65% of the ER-DAG sensor puncta colocalized with Pex30-2xmCherry (Fig. 1d). This finding indicates that some Pex30 subdomains become highly enriched in DAG when LD formation is induced, consistent with the idea that the subdomains are regions where LDs formation occurs when it is induced. It should be noted that it remains possible that

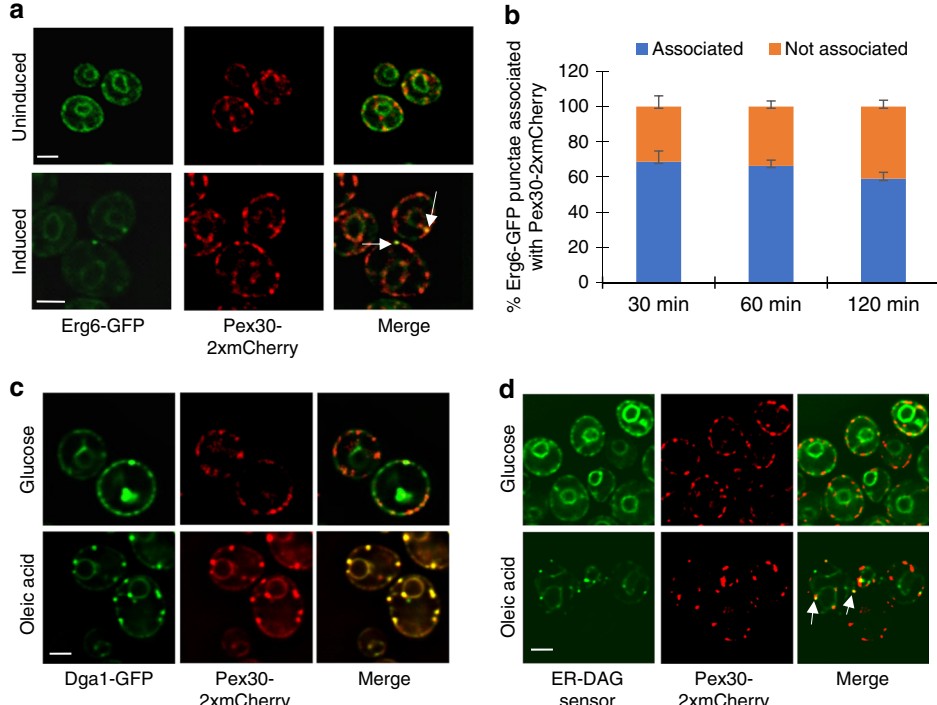

**Fig. 1** Most nascent LDs are associated with Pex30 subdomains. **a** *GAL1-LRO1* 3Δ cells expressing endogenously tagged Erg6-GFP and Pex30-2xmCherry were visualized growing in a medium containing raffinose (uniduced) or 30 min after galactose addition (induced). White arrows indicate association of Erg6-GFP and Pex30-2xmCherry puncta. **b** Percent of Erg6-GFP punctae that are associated with Pex30-2xmCherry after galactose addition. Values are mean +/− s.d. of three independent experiments. **c** Wild-type cells expressing endogenously tagged Pex30-2xmCherry and containing the plasmid Yep181-Dga1-GFP were grown to stationary growth phase in SC medium (glucose). The cells were washed, incubated with fresh SC containing 1 mM oleic acid, and imaged after 1 h (Oleic acid). **d** Wild-type cells expressing endogenously tagged Pex30-2xmCherry and the ER-DAG sensor were grown as in **C**. Bars = 3 μm. White arrows indicate colocalization of Pex30-2xmCherry and ER-DAG sensor punctae

C1a/b-PKD binds TAG as well as DAG, which could contribute to enrichment of the ER-DAG sensor to LD biogenesis sites.

**Pex30 subdomains associate with LD biogenesis proteins**. Since most nascent LDs are associated with Pex30 domains when LD production is induced, we wondered whether LDs remain associated with Pex30 in cells when LD production has not been stimulated. Using Pearson's co-efficient, we determined whether Pex30 colocalized with two proteins known to be at or near sites of LD biogenesis. One such protein is Nem1, which is part of a phosphatase complex that regulates Pah1, a phosphatidic acid phosphatase that is an important regulator of lipid metabolism and TAG production in cells[29–31]. Nem1 forms puncta on the ER that are near sites of LD biogenesis[32,33]. We found that Nem1-GFP punctae partially co-localized with Pex30-2xmCherry (Fig. 2a, d), suggesting these proteins are in close proximity, consistent with the idea that most Pex30 subdomains remain associated with growing and mature LDs. We confirmed Pex30 subdomains and ER exit sites (ERES) are distinct regions of the ER, as a previous study has shown[21]; there was little colocalization of Pex30-2xmCherry and the ERES marker Sec13-GFP[34] (Fig. 2b, d).

We also determined whether Sei1, the yeast homolog of seipin, colocalizes with Pex30 subdomains. Similar to Nem1-GFP, we found that Sei1-GFP puncta partially co-localized with Pex30-2xmCherry domains (Fig. 2c, d); ~40% of the Sei1-GFP were associated with Pex30-mCherry. It may be that the Sei1 puncta away from Pex30 puncta are those not associated with nascent LDs, since a previous study found that only some seipin puncta are associated with growing LDs[35]. To test this, we visualized cells expressing Pex30-2xmCherry, Sei1-GFP, and the LD marker

Erg6-BFP. We found when Pex30-2xmCherry and Sei1-GFP are associated, they were also associated with Erg6-BFP about 95% of the time (Fig. 2e, f). In addition, when Sei1-GFP and Erg6-BFP puncta are associated, about 60% were also associated with Pex30-2xmCherry (Fig. 2e, f). This suggests that when Sei1 is together with Pex30 they are often associated with sites of LD biogenesis.

To estimate how often Pex30 domains remain associated with mature LDs, we determined the percent association of Pex30-2xmCherry and Erg6-GFP (Fig. 2g). We found about 70% of Erg6-GFP puncta were associated with Pex30-2xmCherry, consistent with idea that mature LDs often remain close to Pex30 domains. Taken together, these findings suggest that Pex30 subdomains frequently remain associated with growing and mature LDs.

**LD biogenesis is altered in cells lacking Pex30**. To investigate the role of Pex30 in LD biogenesis, we examined LDs in cells lacking Pex30 (*pex30Δ*). LDs were examined by BODIPY staining (Fig. 3a) and by EM (Fig. 3b–f). LDs in *pex30Δ* cells were often more clustered and smaller than those in wild-type (WT) cells. The decrease in LD size is probably not caused by a change in the level of neutral lipids in *pex30Δ* cells (Fig. 3g, h) or in the rate of neutral lipid synthesis in *pex30Δ* cells (Supplementary Fig. 2A, B). Although these cells had a small but significant decrease in TAG levels, this change is probably not large enough to decrease LD size. The change in LD size in *pex30Δ* cells could be because Pex30 affects membrane surface tension at sites of LD biogenesis. A similar role has been proposed for REEP1, a mammalian reticulon-like ER-shaping protein[36,37].

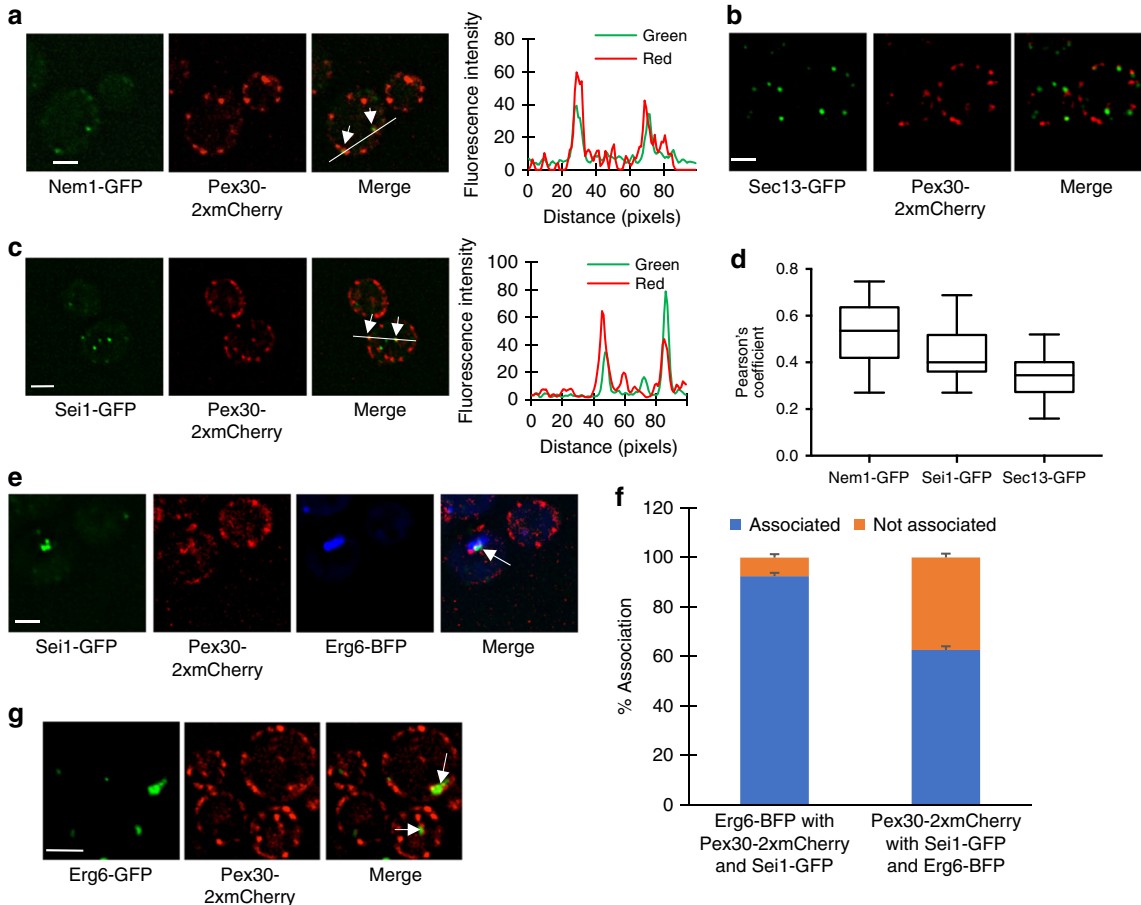

**Fig. 2** Pex30 subdomains associate with LD biogenesis proteins. **a–c** Wild-type cells expressing endogenously tagged Pex30-2xmCherry and Sei1-GFP (**a**), Sec13-GFP (**b**), or Nem1-GFP (**c**) were grown in SC to early stationary growth phase and visualized live. Arrows indicate sites where Pex30-2xmCherry associates with other markers. Graphs to right of **a** and **c** show signal intensity on white line. **d** Pearson's coefficient for colocalization of Pex30-2xmCherry punctae associated with Nem1-GFP, Sei1-GFP, and Sec13-GFP; $n \geq 50$ cells. The box covers the 25th to 75th percentiles, the whiskers mark the highest and lowest values and the median is indicated. **e** Wild-type cells expressing endogenously tagged Pex30-2xmCherry and Sei1-GFP and Erg6-BFP expressed from a 2 μ plasmid were grown in SC to early stationary growth phase and visualized live. Arrow indicates site where Pex30-2xmCherry associates with other markers. **f** Quantification of experiment shown in **e**. Values are mean +/– s.d. of three independent experiments. **g** Wild-type cells expressing endogenously tagged Pex30-2xmCherry and Erg6-GFP were grown in SC to early stationary growth phase and visualized live. Arrows indicate sites where Pex30-2xmCherry and Erg6-GFP associate. Bars = 3 μm. In **a–c** and **g** stacks of three images with a step size of 0.25 μm were deconvolved; images from a single plane are shown

Four proteins in yeast contain RHDs similar to that of Pex30: Pex28, Pex29, Pex31, and Pex32. We determined whether mutants lacking these proteins had changes in LD size, LD clustering, or neutral lipids levels but found they did not, though there was some LD clustering in cells lacking Pex29 (Fig. 3d, h and Supplementary Fig. 3A-H). These findings suggest the functions of Pex30 do not overlap with those of similar RHD-containing proteins.

We found that production of nascent LDs is slower in cells lacking Pex30. To determine the rate of LD production, nascent LD formation was induced in *GAL1-LRO1* 3Δ cells and the same strain also lacking Pex30 (*GAL1-LRO1* 3Δ *pex30Δ*). These strains lack LDs when grown in media with glucose, but begin to produce LDs when shifted to media that contains galactose and lacks glucose. When LD biogenesis was induced, there was a significant delay in the production of LDs in the cells lacking Pex30 (Fig. 3i). We confirmed this finding by inducing LD production by a second method. When cells are in stationary growth phase, the TAG synthase Dga1 is in the ER but when cells are diluted into fresh media and begin to grow, Dga1 relocalizes to regions in the ER where LD biogenesis occurs[23,24,38]. We found that in cells lacking Pex30, there was a significant delay in the re-localization

Dga1 to LDs (Fig. 3j), consistent with the idea that there is a delay in LD production in cells lacking Pex30.

Since LD biogenesis is altered in cells lacking Pex30, we wondered whether membrane protein diffusion between the ER and the surface of LDs is reduced in these cells; this diffusion rate is known to decrease in cells lacking seipin[10,11]. We used fluorescence recovery after photo-bleaching to estimate the rate of Dga1-GFP diffusion from the ER to LDs. LDs that have Dga1-GFP on the surface were bleached and the rate of fluorescence recovery determined. There was no significant difference in the rate of recovery of WT and cells lacking Pex30 (Supplementary Fig. 3I), indicating that ER-LD connections are normal in cells lacking Pex30.

**Genetic interaction of *PEX30* and *SEI1*.** Since seipin has been suggested to localize to sites where LDs are associated with the ER[12,13], we wondered how LD biogenesis would be affected in cell lacking both Pex30 and seipin (Sei1). Surprisingly, these cells (*sei1pex30Δ*) have a substantial growth defect (Fig. 4a). The defect was corrected when the RHD-containing N-terminal 234 residues of Pex30 were expressed in the *sei1pex30Δ* cells, indicating that

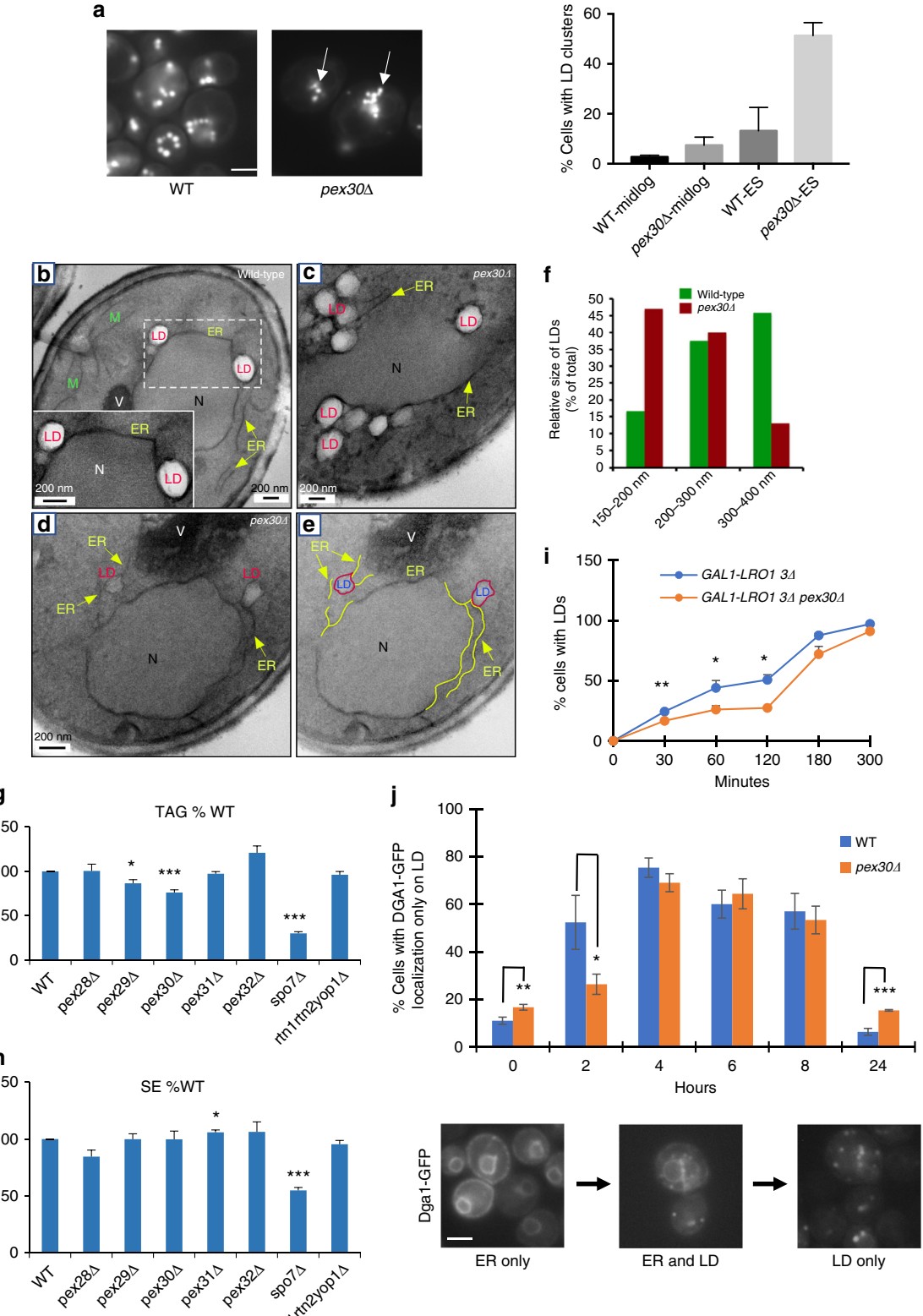

**Fig. 3** Cells lacking Pex30 have altered LDs. **a** Cells in SC were grown to early stationary growth phase and stained with BODIPY to visualize LDs. Stacks of ten images with a step size of 0.2 μm were taken; images from a single plane are shown. Arrows indicate clustered LDs. Scale bar 3 μm. Graph indicates mean +/− s.d. of three independent experiments, *n* = 100 cells; midlog = cells in mid-logarithmic growth phase; ES = cells in early stationary growth phase. **b**–**e** Cells growing in SC were fixed and visualized by EM. Yellow arrows indicate ER. **e** Shows the same image as (**d**), but with portions of the ER highlighted in yellow. **f** Quantification of LD diameter in wild-type and *pex30Δ* cells. *n* = 50 cells. **g**, **h** Cells were grown to early stationary growth phase in SC medium containing [³H] acetate and the relative amount of TAG (**g**) and SE (**h**) determined. **i** Cells were shifted to a medium with galactose (time = 0) and the percent of cells with LDs determined over time; LDs were detected using Erg6-GFP or BODIPY. **j** Cells expressing Dga1-GFP from a plasmid were grown to stationary growth phase in SC (0 h). Cells were washed, resuspended in fresh media, and the percent with Dga1-GFP only on LDs over time determined. Bottom panels show examples of Dga1-GFP distribution patterns: ER only (left), ER and LDs (middle), and LD only (right). **g**–**j** Show mean +/− s.d. of three independent experiments, *p < 0.05, **p < 0.005, ***p < 0.0005, Student's *t*-test. Bars = 3 μm. V vacuole, N Nucleus, LD Lipid droplets

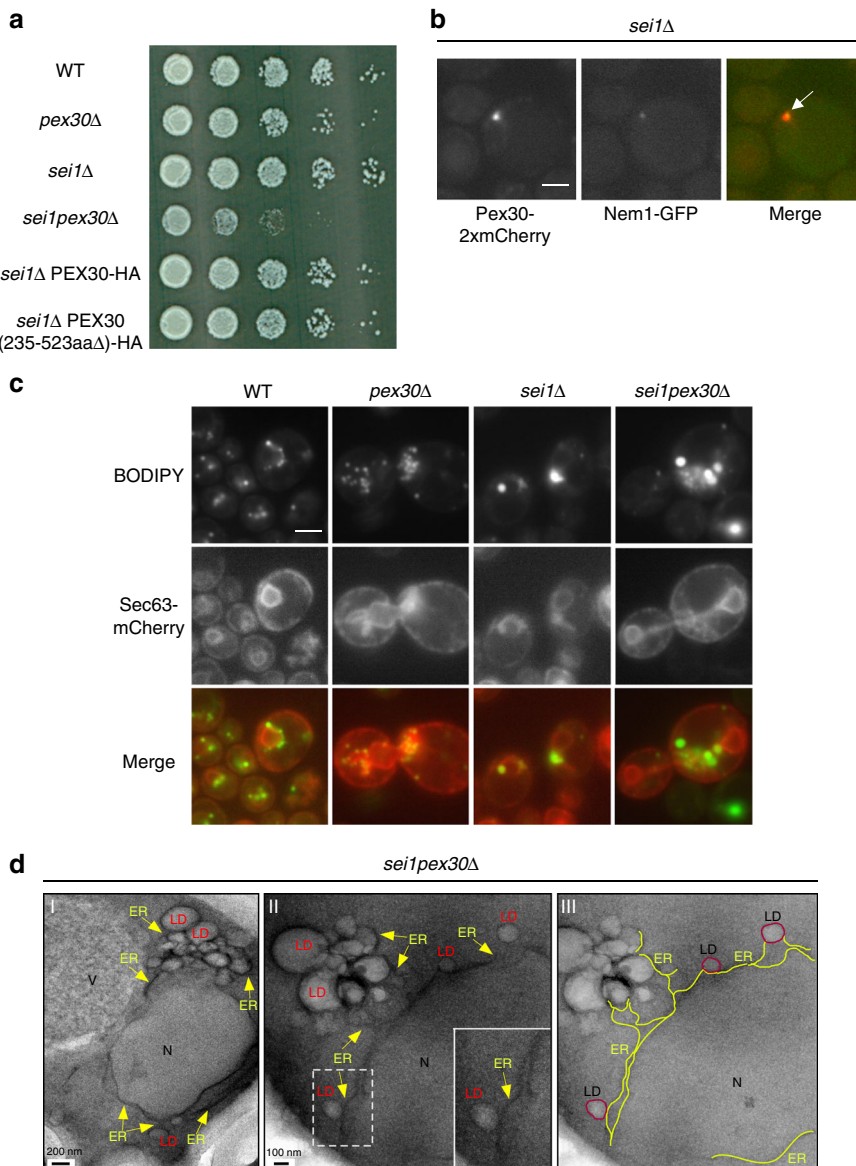

**Fig. 4** Cells lacking Pex30 and seipin have altered LDs and growth defects. **a** Strains were grown to mid-logarithmic growth phase, serially diluted, spotted onto YPD plates, and incubated at 23 °C for 3 days. **b** A *sei1*Δ mutant expressing endogenously tagged Pex30-2xmCherry and Nem1-GFP was visualized live. Arrow indicates colocalization. Bar = 3 μm. **c** Images of cells stained with BODIPY and expressing Sec63-mCherry from a plasmid. Bar = 3 μm. **d** Cells growing in SC without oleic acid were fixed and visualized by EM, abbreviations as in Fig. 3b–e. III shows the same image as II, but with portions of the ER highlighted in yellow

the membrane-shaping function of Pex30 is necessary to support optimal growth of cell lacking seipin (Fig. 4a). Interestingly, elimination of seipin causes a profound redistribution of Pex30 in the ER; Pex30-2xmCherry accumulates in a single punctum that localizes with the LD marker Nem1-GFP (Fig. 4b), indicating that seipin affects the distribution of Pex30 subdomains.

It is not clear why *sei1pex30*Δ cells have a growth defect. We found that *sei1pex30*Δ cells form large clusters of small and large LDs, and the ER associated with the LDs is highly proliferated around the LDs (Fig. 4c, d). These changes could affect the ER function and cause a growth defect. Alternatively, Pex30 and seipin may modulate ER structure or surface tension at LD biogenesis sites, which could affect lipid metabolism. Together, these findings provide additional evidence that Pex30 plays an important role in LD biogenesis and function, and suggest that Pex30 and seipin may have partially overlapping functions.

**MCTP2 has a C-terminal ER-shaping RHD**. Pex30 does not have a mammalian homolog[39] that can be identified using protein BLAST, but we wondered whether there is an RHD-containing protein in higher eukaryotes that plays a similar role. Using the structural homology prediction program HHpred[40], we identified the protein MCTP2 as the closest human homolog of Pex30 (Fig. 5a). MCTP2 is an 878-amino acid ER-resident protein with a membrane-embedded domain related to RHDs near the C-terminus[41,42]. It has four domains: three C2 domains (defined by Pfam) and a domain similar to a Pex24 domain, which is the umbrella name for a domain common to the peroxins Pex24/28/29/30/31/32 (defined by the more sensitive tool HHpred). We have previously shown that the Pex24 domain is related to the hydrophobic region of reticulons, for example sharing a functionally important tryptophan (W684 in MCTP2). Using HHpred, we found that the C-terminal 272 residues of MCTP2

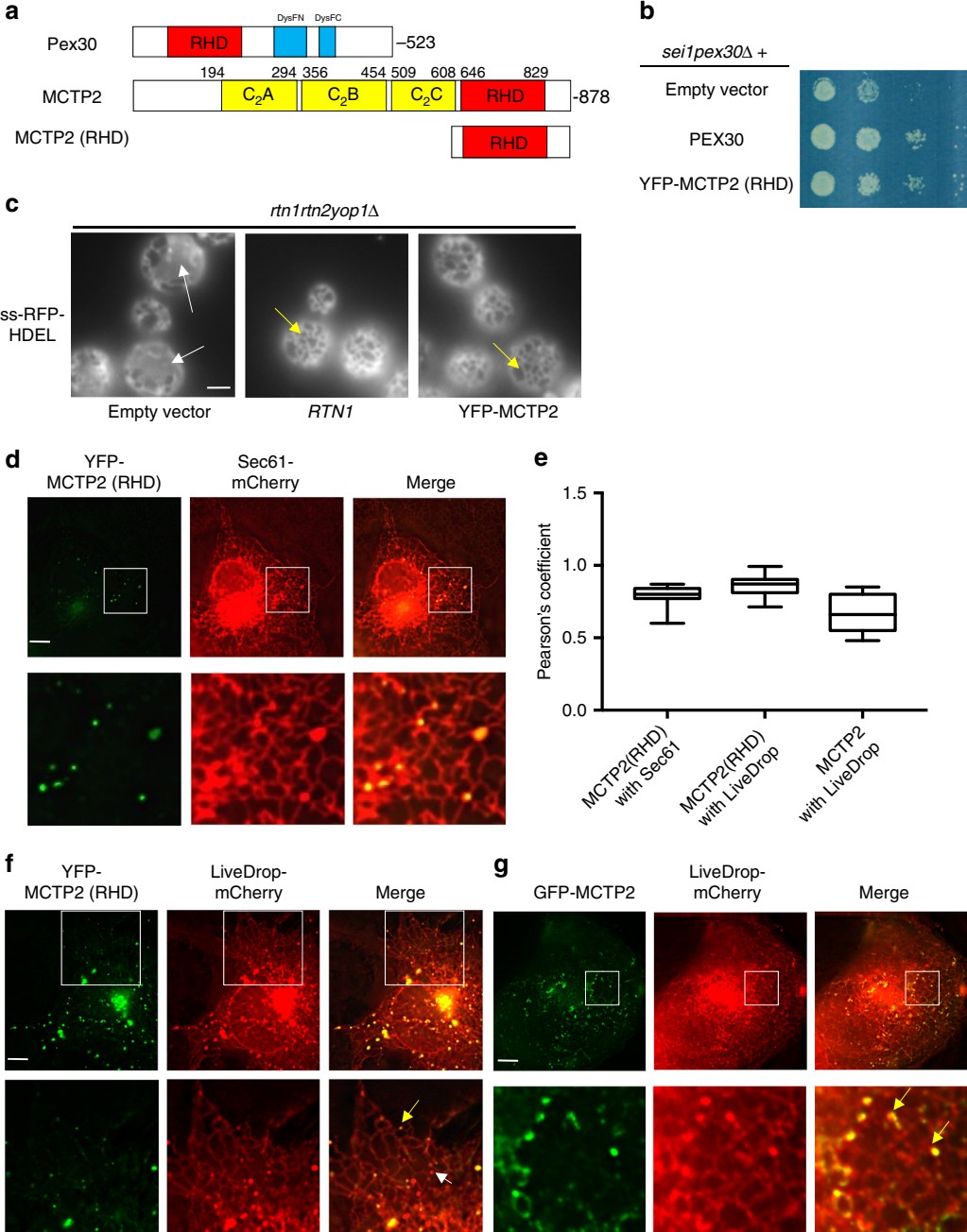

**Fig. 5** Evidence that human MCTP2 is a functional homolog of Pex30. **a** Domains in Pex30, MCTP2, and MCTP2 (RHD). Pex30 contains a dysferlin (DysF) domain that is divided into *N* and *C* portions. **b** Strains were grown to mid-logarithmic growth phase, serially diluted, spotted on to SC plates and incubated at 23 °C for 3 days. **c** Images of the *rtn1rtn2yop1Δ* cells expressing the ER marker ss-RFP-HDEL and Rtn1or YFP-MCTP2 (RHD). Images are of the cell periphery. White arrows indicate ER sheets and yellow arrow show regions containing ER tubules. Bar = 3 μm. **d** Images of COS7 cells co-transfected with plasmids expressing YFP-MCTP2 (RHD) and Sec61-mCherry. **e** Pearson's coefficient of colocalization of experiments shown in **d**, **f**, and **g**. *n* = 100 cells. The box covers the 25th to 75th percentiles, the whiskers mark the highest and lowest values and the median is indicated. **f, g** Images of COS7 cells co-transfected with plasmids expressing LiveDrop-mCherry and YFP-MCTP2 (RHD) (**f**) or GFP-MCTP2 (**g**). Yellow arrow indicates co-localization and white arrow indicates a LiveDrop puncta that does not colocalize with YFP-MCTP2 (RHD) or GFP-MCTP2. In **e**, **f**, and **g**, stacks of 20 images with a step size of 0.3 μm were deconvolved; images from single plane are shown. Bars = 5 μm

(607–878) were related to membrane-embedded portions of the *S. cerevisiae* proteins Pex28-32 and the two reticulon proteins in this yeast (Rtn1 and Rtn2; Supplementary Fig. 4). MCTP proteins are conserved in higher eukaryotes. *Drosophila* and *C. elegans* have one MCTP, whereas humans contain MCTP1 and MCTP2[41].

To determine whether the C-terminal portion of MCTP2 contains an ER-shaping RHD, we expressed the C-terminal 237 amino acids of human MCTP2 fused to YFP, YFP-MCTP2

(RHD), in *S. cerevisiae* under the strong *RTN1* promoter (Fig. 5a). The fusion complements the growth defect of *sei1pex30Δ* cells, as does Pex30, Pex31, and Rtn1 (Fig. 5b, Supplementary Fig. 5A). Cells lacking the reticulons, Rtn1 and Rtn2, and the reticulon-like protein Yop1 (*rtn1rtn2yop1Δ*), have a defect in ER structure[43] that is corrected by overexpression of Pex30[17]. We found that YFP-MCTP2 (RHD) similarly restores ER structure (Fig. 5c). The cortical ER forms large sheet-like structures in *rtn1rtn2yop1Δ*

cells that are not present in WT cells, which contain largely tubular ER in the cortex. We found that cortical ER structure in *rtn1rtn2yop1Δ* cells becomes tubular when YFP-MCTP2 (RHD) is expressed in these cells (Fig. 5c). We previously found that *rtn1rtn2yop1Δ* cells that also lack the lipid regulator Spo7 are not viable but grow when Pex30 is overexpressed[17]. Similarly, overexpression of YFP-MCTP2 (RHD) also rescued the *rtn1rtn2yop1spo7Δ* mutant (Supplementary Fig. 5B). Together, these findings indicate that the C-terminal domain of MCTP2 is an ER-shaping region, probably an RHD, that can functionally replace Pex30 in yeast.

**YFP-MCTP2 (RHD) localizes to sites of LD biogenesis**. We found that YFP-MCTP2 (RHD) localizes to ER subdomains in mammalian cells. YFP-MCTP2 was transiently expressed in COS7 cells together with the ER marker Sec61-mCherry. When YFP-MCTP2 (RHD) was expressed at low levels, it was found in puncta in the ER (Fig. 5d, e). The YFP-MCTP2 (RHD) puncta are stable and remain associated with the same region of the ER over time (Supplementary movie 1). When expressed at high levels, YFP-MCTP2 (RHD) localized all over ER tubules and at the edges of ER sheets (Supplementary Fig. 5C), a localization shared with the reticulons[18] and consistent with the idea that the C-terminal region of MCTP2 contains a RHD.

We next determined whether MCTP2-subdomains are sites of LD biogenesis and associate with LDs. COS7 cells were transiently transfected with YFP-MCTP2 (RHD) and LiveDrop-mCherry, a fusion protein demonstrated to target nascent LDs forming in the ER and mature LDs[10]. We found that most YFP-MCTP2 (RHD) and LiveDrop-mCherry punctae colocalized (Fig. 5e, f). Full-length MCTP2 (GFP-MCTP2) has a similar localization (Fig. 5e, g). LiveDrop-mCherry punctae that did not colocalize with YFP-MCTP2 (RHD) was largely not associated with the ER (Fig. 5f). These findings indicate that MCTP2 localizes to ER sites where new LDs form and suggest that MCTP2 is not associated with mature LDs.

**YFP-MCTP2 (RHD) puncta are not at ERES**. The punctate distribution of YFP-MCTP2 (RHD) puncta throughout the ER suggests that they could be at ERES, where coatomer (COPII) vesicles are generated. We have previously shown that the COPII component Sec23 fused to mCherry (Sec23-mCherry) localizes to ERES in COS7 cells; confirmation that Sec23-mCherry marks ERES) was obtained by co-expressing YFP-PrPC179A, a mutant variant of prion protein, previously shown to move into ERES en route to the Golgi within minutes of ER stress[44]. There was no colocalization of Sec23-mCherry and YFP-MCTP2 (RHD) puncta (Supplementary Fig. 5D), indicating that YFP-MCTP2 (RHD) does not localize to ERES.

**Human MCTP2 and *C. elegans* MCTP play a role in LD biogenesis**. If MCTP2 has a function similar to that of Pex30, depletion of MCTP2 might alter LD size or number. We used siRNA to reduce the expression of MCTP2 in HeLa cells (Fig. 6a) and found a significant decrease in the diameter of LDs (Fig. 6b, c). Depletion of seipin (Fig. 6a) affected LD size (Fig. 6b, c), but the effects of depleting seipin and MCTP2 were not additive. Depletion of MCTP2 in HeLa cells had no effect on peroxisome number (Supplementary Fig. 5E).

We also generated a *C. elegans* strain with a complete deletion of the gene encoding the single MCTP in this animal [*mctp-1 (av112)*]. The number and size of LDs in the intestines of the *mctp-1(av112)* animals were significantly decreased compared to WT animals (Fig. 6d–f). The *mctp-1(av112)* animals had no decrease in embryo viability (Fig. 6g). Together, these findings

suggest that MCTP proteins play a role in LD biogenesis similar to that of Pex30 in *S. cerevisiae*.

**Pex30/MCTP2 sites associate with LDs and peroxisomes/PPVs**. Previous studies have suggested that Pex30 may reside at contact sites with peroxisomes[20]. We wondered whether Pex30/MCTP2 sites in the ER might not only associate with LDs but also with PPVs and peroxisomes, since we previously found that PPVs, are generated at the Pex30 subdomain[17]. YFP-MCTP2 (RHD), LiveDrop-mCherry, and CFP-SKL (a peroxisome marker) were expressed in COS7 cells. About 30% of peroxisomes were associated with YFP-MCTP2 (RHD) that colocalized with LiveDrop-mCherry (Fig. 7a). This association was not random since clockwise rotation of the CFP-SKL image by 90 degrees caused the percent associated to decrease to 7%. Interestingly, peroxisomes are either transiently (Supplementary movie 2) or stably (Supplementary movie 3) associated with the ER subdomains containing MCTP2 and LiveDrop-mCherry.

Similar results were obtained in yeast expressing Pex30-2xmCherry, Erg6-BFP, and the peroxisome marker Pex14-GFP; about 60% of peroxisome were associated with Pex30 domains and LDs (Fig. 7b,c). To visualize PPVs in *S. cerevisiae*, we used strains that lack the proteins Pex3 and Atg1 (*pex3atg1Δ*). The PPVs contain Pex14-GFP and there are typically one or two PPVs per cell. We found that some Pex14-GFP puncta are on vesicles, while others are on the ER, at Pex30 subdomains[17]. To colocalize PPVs, LDs, and Pex30 subdomains, we expressed Pex14-GFP, Pex30-2xmCherry, and the LD marker Erg6-BFP in *pex3atg1Δ* cells. Remarkably, most Pex14-GFP puncta are closely associated or colocalize with Pex30 subdomains and LDs (Fig. 7b, c). These Pex14-GFP puncta could either be present on the ER membrane or PPVs. The association between PPVs and LDs was even more pronounced in *pex3atg1Δ* cells that also lack seipin (Supplementary Fig. 6). This is probably due to the redistribution of Pex30 to a single punctum in *sei1Δ* cells (Fig. 4b). Altogether, these results suggest that LDs and peroxisomes or PPVs are associated with the same Pex30 ER subdomain over time.

## Discussion

In this study, we demonstrate that conserved ER subdomains with specialized RHD-containing proteins are sites where nascent LDs mature. Consistent with this, we found that these sites, in *S. cerevisiae*, are enriched in the TAG precursor DAG and are often associated with Nem1 and Sei1, proteins previously shown to be enriched at sites of LD biogenesis[13,32,33]. The RHD-containing protein at LD biogenesis sites in *S. cerevisiae* is Pex30; and we show that MCTP2 in higher eukaryotes also has an RHD and localizes to ER subdomains containing nascent LDs.

The early steps of LD biogenesis and how sites of LD biogenesis in the ER are determined is not well understood. One question has been whether LDs are generated at defined zones in the ER and whether new sites are generated when LD biogenesis is induced. Since Pex30/MCTP2 sites are relatively stable and exist before LD biogenesis is induced, we propose that they are sites where proteins that mediate LD production can assemble when LD biogenesis is stimulated. Whether the earliest steps of LD biogenesis, probably the formation of small (<50 nm) lenses of neutral lipid in the ER bilayer, occur at these sites remains to be determined. It is more likely that nascent LDs formation occurs throughout the ER and the nascent LD lenses diffuse in the ER to Pex30/MCTP2 sites where they grow and eventually emerge into the cytoplasm. Although Pex30/MCTP2 sites seem to facilitate LD growth, we find they are not required since some LDs mature outside of these sites and elimination or depletion of Pex30 or MCTP2 does not block LD biogenesis. An important question for

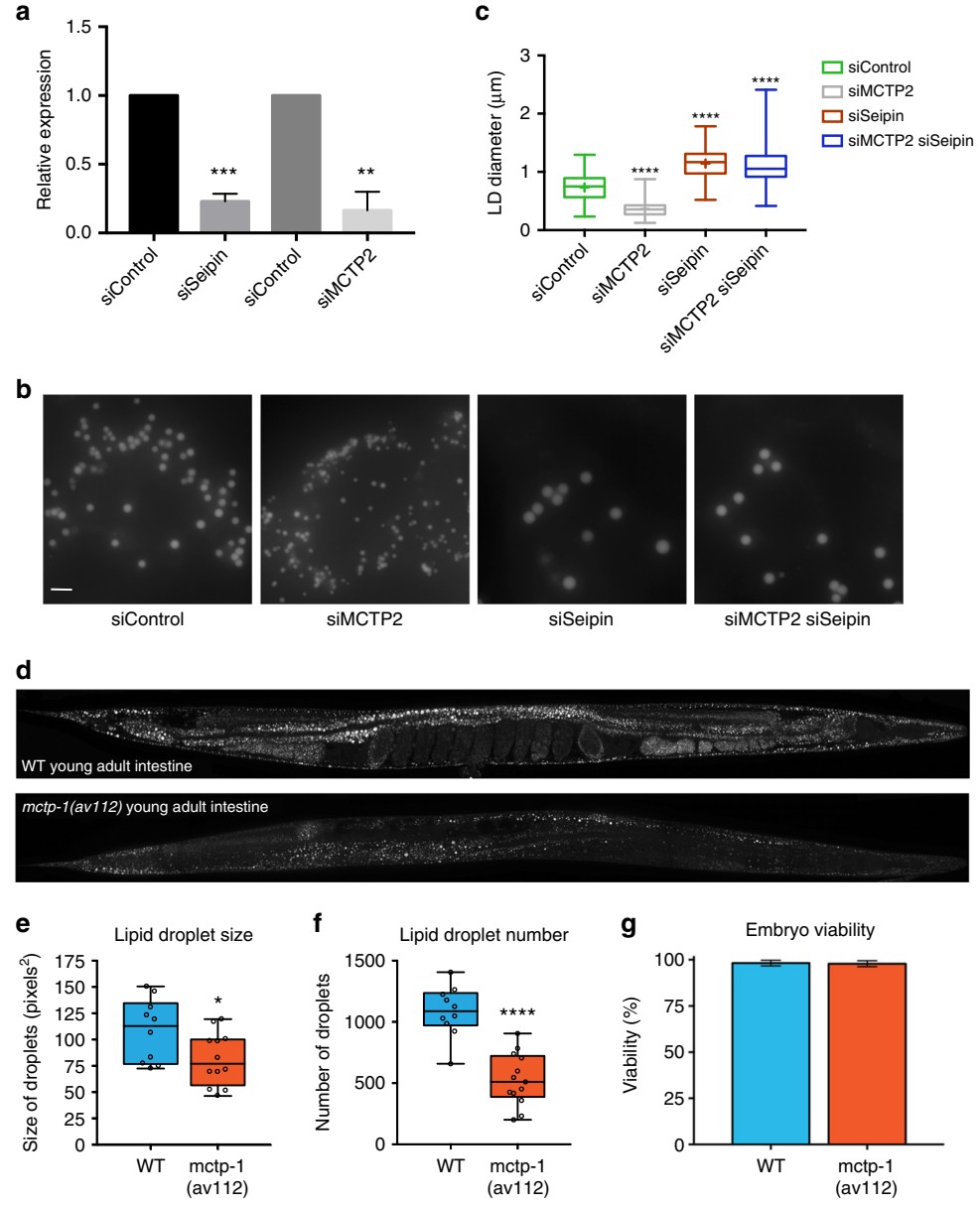

**Fig. 6** Human MCTP2 and *C. elegans* MCTP play a role in LD biogenesis. **a** Knockdown of expression of expression of MCTP2 and Seipin genes in HeLa cells quantified by real-time PCR (mean +/− s.d, $n = 4$ from two independent experiments). **b** HeLa cells stained with BODIPY after incubation with siRNA. Bars = 5 μm. **c** Quantification of diameter of LDs (μm). mean +/− s.d, from two independent experiments. **** $p < 0.0001$. The box covers the 25th to 75th percentiles, the whiskers mark the highest and lowest values and the median is indicated. (**d**) BODIPY staining of WT and *mctp-1(av112)* worms. Images are collapsed Z stacks by maximum projection. **e**–**g** Quantification of LD size (**e**), LD number (**f**), and embryo viability (**g**) of WT and *mctp-1(av112)* animals; * = $p = 0.0326$ and **** = $p < 0.0001$. For **e** and **f**, the box covers the 25th to 75th percentiles, the whiskers mark the 5th and 95th percentiles, the median is indicated, and the measurement from individual worms are overlaid as open circles

the future is to determine how the protein and lipid composition of Pex30/MCTP2 domains differ from the rest of the ER and what role the RHD plays in establishing these domains.

Our findings suggest that, in *S. cerevisiae*, most mature LDs remain associated with Pex30/MCTP2 domains. MCTP2 domains may similarly remain associated with some mature LDs in mammalian cells. In *S. cerevisiae*, LDs do not completely detach from the ER[23] and our findings suggest that mature LDs remain associated with the ER at Pex30 subdomains. The connections between mature LDs and Pex30 subdomains may allow these domains to regulate mature LDs in addition to playing a role in nascent LD formation.

We have previously found that PPVs may be generated at Pex30 subdomains[17], which suggests that these subdomains play a role in both peroxisome and LD biogenesis. Although this is surprising, it has previously been found that Pex3 and Pex19, which are required for peroxisome biogenesis, also insert membrane-embedded proteins into the surface of LDs at ER subdomains[22]. Perhaps there are other proteins that play a role in the biogenesis of both organelles at Pex30/MCTP2 sites. It remains unknown whether the same Pex30 or MCTP2 site can simultaneously give rise to both organelles.

Interestingly, individual Pex30/MCTP2 subdomains often remain associated with both mature LDs and peroxisomes. This

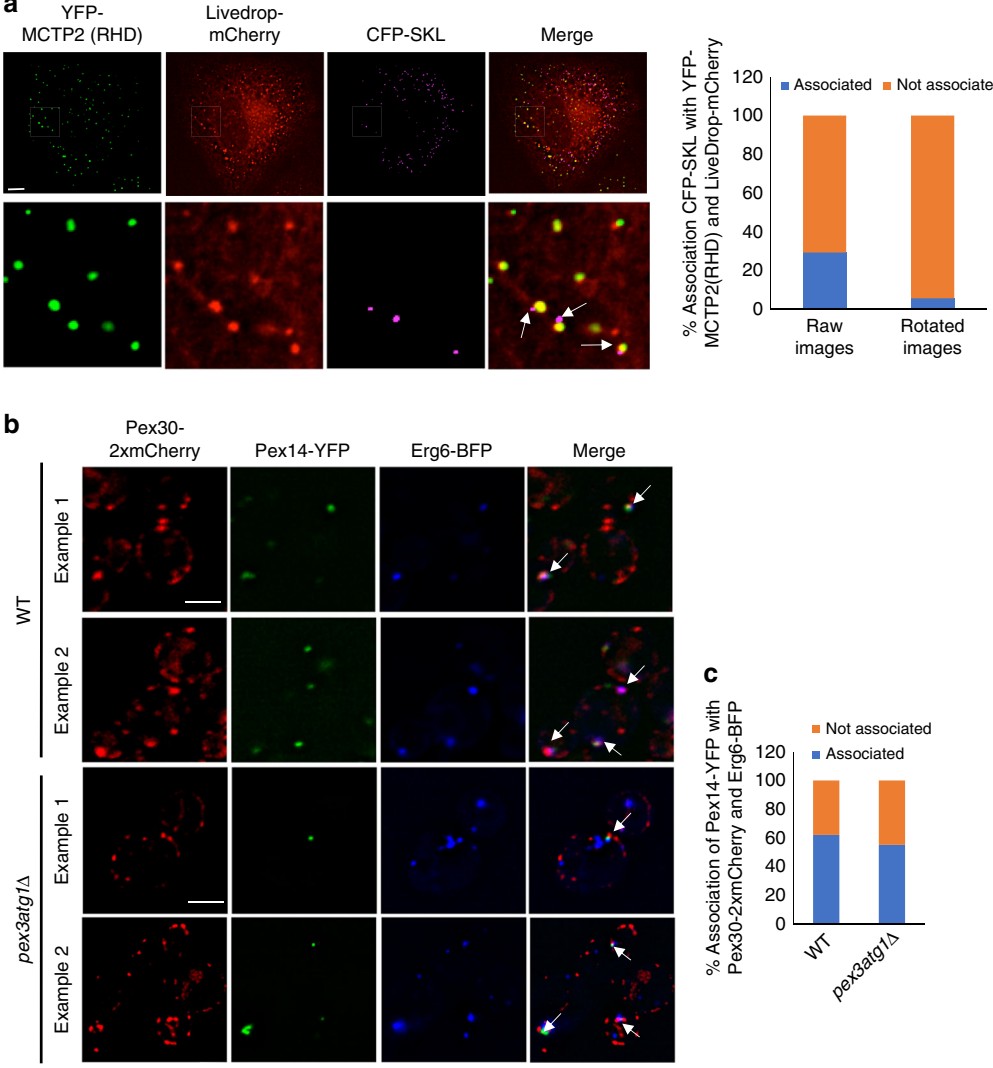

**Fig. 7** Pex30/MCTP2 sites simultaneously associate with LDs and Peroxisomes/PPVs. **a** Images of COS7 cells transiently co-transfected with plasmids expressing YFP-MCTP2 (RHD), LiveDrop-mCherry, and CFP-SKL (a peroxisome marker). Stacks of 20 images with a step size of 0.3 μm were deconvolved; images from single plane are shown. Bar = 5 μm. Graph shows percent association of CFP-SKL puncta associated with YFP-MCTP2(RHD) and LiveDrop-mCherry, from raw images or where the CFP-SKL images was rotated 90 degrees, n = 300 CFP-SKL punctae. **b** Images of WT and *pex3atg1Δ* cells expressing endogenously tagged Pex30-2xmCherry and Pex14-YFP (a PPV marker) and Erg6-BFP from plasmids. Stacks of three images with a step size of 0.25 μm were deconvolved; images from single plane are shown. Bar = 3 μm. **c** Quantification of experiments in **b** indicating percent association of Pex14-YFP punctae with Pex30-2xmCherry and Erg6-BFP, n = 100 cells

finding is consistent with earlier work that suggests Pex30 is present at sites of ER-peroxisome contact. Whether the organelles are associated because they remain in contact with the ER-domain where they were generated remains to be determined. Our finding that PPVs also often remain associated with LDs and Pex30 subdomains suggests that PPVs and perhaps mature peroxisomes remain near their site of origin. As LDs and peroxisomes are also known to make close contacts[45,46], Pex30/MCTP2 subdomains may facilitate intracellular signaling between the ER, peroxisomes, and LDs.

## Methods

**Yeast strains and plasmids**. The strains and plasmids used in this study are listed in Tables S1 and S2. Deletion strains were constructed by mating or PCR-based targeted homologous recombination to replace the ORF of genes of interest with cassettes (Longtine et al., 1998). PCR-based targeted homologous recombination was also used to generate endogenously expressed C-terminally tagged fusion proteins. The 2xmCherry-URA3 and GFP-HIS3MX6 cassettes were obtained from O. Cohen-Fix (National Institutes of Health/National Institute of Diabetes and

Digestive and Kidney Diseases, Bethesda, MD), YFP-KanMX6 from J. Cooper (National Institutes of Health/National Cancer Institute, Bethesda, MD), and yEmCherry-HIS5MX6 and pRS305-PHO8-3xBFP from J. Nunnari (University of California, Davis, Davis, CA).

The plasmid encoding ER-DAG sensor was constructed by fusing the portion of the human protein kinase D gene encoding amino acids 136–343 (obtained from Tamas Balla, National Institute of Child Health and Human Development, NIH) to genes encoding GFP and the tail-anchored transmembrane domain of *S. cerevisiae* Ubc6 under the ADH1 promoter in the plasmid YEplac181. The plasmids used in live-cell imaging of COS7 cells were Sec61-mCherry from T. Rapoport (Harvard University), YFP-MCTP2 from T. Sudof (Stanford University), LiveDrop-mCherry from T. Walther (Harvard University), Sec23-mCherry from P. Sengupta (Janelia Research), and CFP-SKL and RFP-SKL from J. Lippincott-Schwartz (Janelia Research). The full-length GFP tagged MCTP2 was cloned into pEGFP-C1 plasmid at Xho1/KpnI restriction sites using CloneEZ PCR cloning kit (catalog # L00339) from Genscript.

**Media and growth conditions**. Yeast cells were grown at 30 °C, unless otherwise indicated, in YPD medium (1% Bacto yeast extract, 2% Bacto Peptone, and 2% glucose) or in synthetic complete (SC) media containing 2% glucose, 0.67% yeast nitrogen base without amino acids (United States Biological), and an amino acid

mix (United States Biological). In some cases the glucose in SC was replaced with 2% raffinose or 2% galactose. When inducing LD production, cells were washed with sterile water twice and transferred to SC containing 1 mM oleic acid and 1% Brij58. Images were taken within 1–2 h.

COS7 cells (ATCC, CRL-165) were cultured in DMEM (Gibco) supplemented with 10% FBS (Gibco) and 2 mM L-glutamine (Gibco) at 37 °C in humidified air containing 5% $CO_2$. Prior to live-cell imaging, the medium was replaced with $CO_2$-independent medium (Gibco) containing 10% FBS and 2 mM L-glutamine.

**Generation of *mctp-1(av112)* C. elegans**. *mctp-1(av112)*, which has complete deletion of the gene encoding MCTP, was generated by following the non-cloning Co-CRISPR conversion technique[47] using *dpy-10* as the co-conversion marker[47]. Strains were cultured using standard conditions.

**siRNA treatment and real-time quantitative PCR**. Specific sets of 4 siRNAs to silence human MCTP2 (MU020810010002) and Seipin (MU016749000002) were purchased from Dharmacon. The negative control siRNA was obtained from Thermofisher (Catalog # 4390843). For siRNA transfections, Hela cells were plated at 50% confluency in six-well plates and left overnight at 37 °C in DMEM along with 10% FBS. The next day, cells were transfected with 50 pmol control, MCTP2 or Seipin siRNA using DharmaFECT transfection reagent according to the manufacturer's instructions. 40 h posttransfection, cells were used for real-time quantitative PCR or live-cell imaging.

Total RNA was isolated from control, MCTP2, Seipin, and both MCTP2 and Seipin siRNA–treated Hela cells using PureLink RNA Mini kit (Ambion) according to the manufacturer's instructions. cDNA was synthesized from 1.5 µg total RNA with qScript cDNA SuperMix (Quanta Biosciences) in 20 µl reaction volume according to the enzyme supplier's instructions. Quantitative real-time PCR was performed in a 10 µl reaction mixture containing 1 µl cDNA, primers, and SYBR Green mix (FastStart Universal, Roche). The sequences of primers used are, GAPDH (5′ CTTCGCTCTCTGCTCCTCCTGTTCG 3′, 5′ ACCAGGCGCCCAAT ACGACCAAAT 3′) MCTP2 (5′ CCAGTGGGAATCCACATTAAGA, TGTACC GCAGTGGAATGAAATA 3′), Seipin (5′ TTCCTCTATGGCTCCTTCTACT 3′, 5′ GACCAAGAACATGCCCAAATC 3′). The enzyme was activated at 95 °C for 20 s. After activation, the reaction mixture was amplified for 40 cycles under the following conditions: denaturing for 1 s at 95 °C and annealing and extension for 20 s at 60 °C. Real-time PCR analysis was done on Real-Time PCR system (Applied Biosystems). Gene expression was normalized to that of GAPDH and data are presented as the "fold change" relative to the corresponding siRNA for control, MCTP2 and Seipin according to the 2-ΔΔCT (change in cycling threshold) method.

**BODIPY staining in yeasts, HeLa cells, and C. elegans**. When staining LDs with BODIPY, yeasts cells in early stationary growth phase were washed with phosphate buffered saline and incubated with 0.5 µg/ml BODIPY 493/503 (Invitrogen) for 10 min.

To induce LDs, Hela cells were incubated with 300 µM BSA-oleic acid (Sigma # O3008) for 14–18 h, stained with 5 µg/ml BODIPY for two h and washed with PBS. Cells were shifted to $CO_2$-independent medium (Gibco) containing 10% FBS and 2 mM L-glutamine before live-cell imaging.

Worms were age matched to 23 h post-L4 stage, 10–15 worms were incubated in BODIPY493/503 (Invitrogen) at 6.7 ug/mL in M9 Buffer for 20 min, followed by 3 wash cycles in M9 Buffer. They were immediately mounted in M9 Buffer for confocal imaging on a Nikon (Garden City, NY) E800 spinning disk confocal microscope using MetaMorph imaging software.

**LD quantification**. The Z-stacks images of LDs in Hela cells were collapsed using maximum projection. The LD diameter (µm) was measured using "measure distance" tool (Softworx, Applied Precision Ltd.). In *C. elegans*, imaging experiments were repeated a minimum of three times. Using FIJI, Z-stacks were selected to cover most of the worm intestine and collapsed by maximum signal projection The intestine was cropped out of these images and subjected to a Macro that sets a threshold for the image at 70 intensity units using the "MaxEntropy" setting, makes the image binary, performs a watershed calculation to isolate overlapping signals, and then uses the "Analyze Particle" function to count and measure individual signals (bounds: size 20-Infinity pixels[2], circularity 0.2–1.0). All the values were recorded using Microsoft Excel and analyzed in Prism 7.0a.

**Fluorescence microscopy**. For Fig. 3, Fig. 5c, and Supplementary Fig. 3, cells were imaged live in growth media using a BX61 microscope (Olympus) with a UPla-nAPO Å~100/1.35 lens and a Retiga EX camera (QImaging) and processed using iVision software (version 4.0.5). For Figs. 1, 2, 4, 5D, 5F, 5G, 6, and 7 and Supplementary Figs. 1, 5, and 6, imaging was performed at 30 °C in an Environmental Chamber with a DeltaVision Spectris (Applied Precision Ltd.) comprising a wide-field inverted epifluorescence microscope (IX70; Olympus), a 100 Å~ NA 1.4 oil immersion objective (UPlanSAPO; Olympus), and a charge-coupled device Cool-Snap HQ camera (Photometrics). For COS7 cells imaging, cells were cultured in MatTek 35 mm petri dish, 14 mm microwell, No. 1.5 coverglass, (0.16–0.19 mm). Cells were transfected with indicated plasmids using Lipofectamine 2000

(Invitrogen) according to the manufacturer's instructions. Time-lapse images were acquired every 60 s for 10 min for Supplementary Fig. 2 and every 12 s for 2.5 min for Supplementary Fig. 3. Images were deconvolved (conserved ratio method) and Pearson's coefficient was measured using SoftWorx (Applied Precision Ltd.). Brightness and contrast were adjusted using Photoshop CC (Adobe Systems).

**Electron microscopy (EM)**. Yeast cells were grown to mid-logarithmic growth phase, and 10 $OD_{600}$ units of cells were pelleted and fixed in 1 ml of fixative media (2.5% glutaraldehyde, 1.25% PFA, and 40 mM potassium phosphate, pH 7.0) for 20 min at room temperature. Cells were pelleted, resuspended in 1 ml fresh fixative media, and incubated on ice for 1 h. The cells were pelleted, washed twice with 0.9% NaCl, once with water, incubated with 2% $KMnO_4$ for 5 min at room temperature, centrifuged, and resuspended in 2% $KMmO_4$ for 45 min at room temperature for en-bloc staining[48]. The cells were dehydrated using ethanol, embedded using Spurr's resin (Electron Microscopy Sciences), and polymerized. Semi- and ultrathin sections were produced with a diamond knife (DiATO ME) on an ultra-microtome (Ultracut UCT; Leica Biosystems), collected on 200 mesh copper grids (Electron Microscopy Sciences), poststained with uranyl acetate and lead citrate, and visualized with a Tecnai T12 transmission electron microscope (FEI), operating at 120 kV. Pictures were recorded on a bottom-mounted 2k Å~ 2k CCD camera (Gatan). Brightness and contrast were adjusted to the entire images using Photoshop (version CC 2014).

**Neutral lipids detection**. To measure the rate of neutral lipid synthesis, strains were grown to 1 $OD_{600}$ per ml, 10 µCi/ml [$^3$H] acetate (American Radiolabeled Chemicals) was added. Cells (10 $OD_{600}$ units) were harvested at indicated times. To measure the neutral lipids at steady state, 10 $OD_{600}$ units of cells were harvested at early stationary growth phase in SC glucose containing 10 µCi/ml [$^3$H] acetate. To extract neutral lipids, cells were lysed using a Precellys24 homogenizer, and lipids were extracted[49]. To quantitate TAG and SE, the lipids were spotted onto silica gel 60 TLC plates (EMD Millipore) and developed with hexane-diethylether-acetic acid (80:20:1). Lipids on TLC plates were quantified with a RITA Star Thin Layer Analyzer (Raytest).

**Dga1-GFP localization assay**. Cells expressing Dga1-GFP from the *DGA1* promoter in the centromeric plasmid YCplac111 were grown in SC. These cells were diluted in fresh medium and incubated for 24 h followed by dilution to 0.3 $OD_{600}$ units/ml. These cells were then imaged at the indicated times to determine the localization of Dga1-GFP.

**Data availability**. The authors declare that the data supporting the findings of this study are available within the paper and its supplementary information files.

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

## Acknowledgements

This work was supported by the Intramural Research Program of the National Institute of Diabetes and Digestive and Kidney Diseases. T.P.L. was supported by the Biotechnology and Biological Sciences Research Council Bioinformatics and Biological Resources Fund (grant BB/M011801). We thank T. Rapoport, T. Sudof, T. Walther. T. Balla, P. Sengupta, and J. Lippincott-Schwartz for providing reagents, J. Cooper for use of her Deltavision microscope and T. Balla and Y. Ye for critically reading the manuscript.

## Author contributions

A.S.J. and W.A.P. designed the experiments. V.C. performed the EM. B.N. and A.G. performed experiments with C. elegans. P.S.-K. performed the experiments with the ERES marker Sec23-mCherry. All other experiments were performed by A.S.J. T.P.L. performed the HHpred analysis. A.S.J. and W.A.P. analyzed the data and wrote the manuscript.
