## [Peer Review File · Nature Communications]

Reviewers' comments:

Reviewer #1 (Remarks to the Author):

Lipid droplets are ER-derived neutral lipid storage organelles that play central roles in metabolic diseases. Although emerging data implicate Sei1 and the FIT proteins in lipid droplet biogenesis, the mechanisms involved remain almost completely unknown. In this manuscript, the authors discover a fundamental link between lipid droplet and peroxisome biogenesis. This is particularly intriguing given the metabolic connections between these two organelles, with fatty acid storage in lipid droplets and fatty acid breakdown in peroxisomes. The data presented demonstrate that Pex30 and Sei1 localize to a subdomain of the ER and play partially redundant roles in lipid droplet and peroxisome biogenesis; the double knockout strains have dramatic defects in both processes. There is not much advance into the mechanism of Pex30 and Sei1 in organelle biogenesis. An important contribution from this manuscript is the identification of the putative mammalian ortholog of Pex30, but unfortunately this aspect of the manuscript is not fully explored.

The identification of Pex30 in lipid droplet biogenesis as well as the connections between lipid droplet and peroxisome biogenesis are important. However, a more complete characterization of the putative mammalian ortholog is necessary.

Major comments

- One of the most exciting aspects of this manuscript is the identification of a putative Pex30 mammalian ortholog. However, only some preliminary experiments are performed. Other than co-localization of an overexpressed small fragment of YFP-MCTP2 with LiveDrop, there is little characterization of MCTP2 in mammalian cells. Does full length MCTP2 and endogenous MCTP2 localize to these ER subdomains? What happens to lipid droplets and peroxisomes following deletion of MCTP2 or MCTP2 and MCTP1? Is there a similar relationship with seipin? What happens to lipid droplets and peroxisomes in an MCTP2 and seipin double delete? These experiments are essential.
- In yeast, the lipid droplet membrane and the outer leaflet of the ER are contiguous, allowing exchange of proteins. This has been shown before using FRAP analysis of fluorescently tagged Dga1 and other lipid droplet proteins. The authors propose an interesting possibility that Pex30 may impact ER-LD contacts, but examine this possibility using an unconventional assay involving measurement of the percentage of cells with Dga1-GFP in the ER or in lipid droplets at various growth phases. The localization of Dga1-GFP to these compartments could be affected by many variables and it is not clear that this accurately reflects exchange at ER-lipid droplet contacts. The movement of Dga1-GFP between the compartments should be measured by FRAP in the WT and mutant strains.
- Are Pex30 and Sei1 (and Nem1) at the same sites in the ER? In the fluorescence microscopy images in Figure 1E/F they seem to be next to each other?
- Is it surprising that DAG builds up at sites of lipid droplet biogenesis given that DAG seems to inhibit lipid droplet budding in vitro (M'barek et al. Dev Cell 2017)?
- Some additional speculation about the mechanism by which Sei1 and Pex30 act to promote organelle biogenesis in the conclusion section would be helpful. Why would this specific reticulon be important? Are the defects in biogenesis related to aberrant ER morphology similar to disruption of atlastin?

Minor comments

1. "the Kapito group..." should be "the Kopito group..." (line 69)
2. "when the cells were grow in..." should be "when the cells were grown in..." (line 124)

Reviewer #2 (Remarks to the Author):

Joshi and colleagues propose that a conserved ER subdomain serves as a shared site for the biogenesis of lipid droplets and peroxisomes. They study the localization of Pex30 and proteins involved in lipid droplet biogenesis in yeast and show that Pex30 has a negative genetic interaction with Sei1. In the *pex30delta/sei1delta* mutant, lipid droplets cluster together in the perinuclear region of the cell. The authors then switch to characterize a mammalian protein MCTP2, which they claim contains a reticulon homology domain that is similar to Pex30. This domain is able to tubulate ER membranes in yeast, and localizes to foci in the ER that also overlap with a lipid droplet reporter. A final figure presents colocalization experiments in yeast and mammalian cells with markers for Pex30 (or MCTP2), peroxisomes, and lipid droplets to suggest that these organelles are formed from the same subdomains of the ER.

Overall, the data presented (and the presentation itself) are not sufficiently well-developed for publication at this stage. Most of the experiments are descriptive in nature, are performed suboptimally, lack quantification, and can be interpreted differently. Major concerns are outlined below:

1. The ostensible reason the authors investigate a role for Pex30 in lipid droplet biogenesis is that they previously showed “10-fold more Pex30 domains than there are PPVs in cells.” They “previously identified ER subdomains in *S. cerevisiae* that contain Pex30” and there are “~25 Pex30-containing puncta in the ER per cell.” However, this misrepresents scientific record. It was Yan et al., MBoC, in 2008, who showed that Pex30 exists in ER subdomains, and this was further defined by Davids et al., 2013 and Mast et al., 2016 to reveal that there are ~10-fold greater Pex30 domains than peroxisomes, and that these domains can serve as sites for regulating peroxisome biogenesis, both as contact sites for peroxisomes, and also as sites for regulating preperoxisomal vesicle budding from the ER. Only Davids et al., is cited in this manuscript and unfortunately not in a proper context here. This is not the only example. The authors should review the citations carefully to ensure they are appropriate throughout the manuscript.
2. There are no controls to show that the Erg6-GFP foci in figure 1 correspond to de novo formed lipid droplets.
3. The quantification in figure 1B lacks error bars, and since it is quantifying only those Erg6-GFP foci that have been captured in a single-focal plane through the center of the cell, how reflective is this of the majority of Pex30 foci, which are cortically localized? What would colocalization with other ER markers look like? For example, would Rtn1 which is known to interact with Erg6 have greater or less colocalization than Pex30? What about Sec61?
4. In figure 1C and throughout this manuscript, Pex30-2xmCherry looks mislocalized to the general tubular ER and not confined to punctate subdomains. How do the authors know that the 2x mCherry tag isn't altering the dynamics and function of Pex30? This mislocalization clearly alters the interpretation of the colocalization between Pex30 and lipid droplet markers.
5. Given that the authors are relying on a single focal plane from a wide field microscope for most of their experimental work, how are they able to distinguish between objects that truly colocalize and those that are merely adjacent to one another within the ER and between the ER and other organelles? The line tracings shown for Sei1 and Nem1 suggest that these proteins are not colocalized but sit adjacent to each other. How is this unique for Pex30 and not a general feature of all ER membrane proteins?

6. The authors conduct no standard statistical tests (Pearson's, Mander's, etc.) nor do they perform any rigorous object based colocalization analysis, which at a minimum, should be performed for all colocalization analyses in the manuscript.

7. The authors selectively cite Pagac et al., 2016 to claim that Pex30 physically interacts with Sei1, and yet from the experimental method in that manuscript there can be no claim of specificity, or direct interaction. This claim must be established experimentally as part of this work.

8. The authors have developed a fluorescent "DAG sensor" and yet fail to provide any evidence that it binds DAG, and whether that binding is exclusive to DAG. Until then, these experiments are uninterpretable.

9. Lipid droplets and peroxisomes are dynamic organelles that respond to changes in cell state and media conditions making comparisons between results from multiple growth conditions difficult to interpret. For example, in figure 2, cells grown to early stationary phase and labeled with BODIPY are compared to cells from mid logarithmic phase and processed for EM. These data should be generated under the same conditions, quantified, and images presented should be representative of the dominant phenotype.

10. The claim that MCTP2 has a reticulon homology domain similar to Pex30 must be taken at face value as no structural or evolutionary data are presented to verify this claim. Similarly, the authors state that there is no mammalian homologue of Pex30 but do not cite or provide evidence in defense of this claim. These analyses must be presented for critical evaluation.

11. The experimental evidence to support the authors claim that MCTP2 is a functional ortholog of Pex30 is weak. First, the authors show that expressing the RHD domain of MCTP2 tubulates ER membranes in the reticulon deletion strain, as they showed for Pex30 previously, and that expressing this RHD domain in the pex30/sei1 double mutant restores growth. But this doesn't demonstrate that MCTP2 is the functional ortholog of Pex30, it merely demonstrates that the RHD of MCTP2 can tubulate membranes. Does expressing the RHD domain of Pex31, or Rtn1, or any other RHD domain have a similar effect?

12. The experiments with the YFP-MCTP2 fusion protein in COS7 cells are difficult to interpret. Overexpression of a tubulating protein will likely have dramatic pleiotropic effects and yet the authors perform no control experiments. In figure 4 and 5 YFP-MCTP2 is punctate, and yet in supplemental figure 3 it is all over the tubular ER. How is the association between this fusion protein and the LiveDrop or PTS reporters unique or functionally relevant? There is too much speculation about the role of this protein.

13. The example in yeast presented in figure 5a, is unconvincing. There is no quantification and even if there were what conclusion could be drawn? These cells have no peroxisomes and are deficient in autophagy. The Pex14 foci may be a PPV or it may an attempt by the cell to sequester PMPs away from other membranes to prevent unwanted affects. Is the ER normal in these cells? Is the UPR upregulated?

Reviewer #3 (Remarks to the Author):

In this manuscript, Joshi et al. find that the same subdomains of the ER give rise to two morphologically and functionally different organelles. They show that these subdomains are defined by

the reticulon-like protein, Pex30 and find a similar role for the mammalian homologue MCTP2.

This is a very interesting study and it gives us novel insights into organelle biogenesis from the ER.

Major points

-The claim that Pex30 recognizes DAG-enriched domains is not properly supported. In Figure 1H, the authors present a biosensor for ER DAG. Its robustness should be validated further. Could it bind TAGs as well? It is important to validate that the signal obtained is specific, for example, by including a C2 domain mutant.

Additionally, the lipid strip experiment in Supp Fig 1D. hardly supports the claim that Pex30 could be regulated by DAG accumulation. Pex30 seems to bind PC or the "control" (whatever it is) better. The authors should clarify this and specify what the "control" is. Are there any other non-labelled lipid species on the strip?

Altogether, I think that all mention of DAG should be removed unless better validation is provided for both the biosensor and the lipid strip experiment.

-In Figure 1E, the authors conclude that Sei1 puncta not co-localizing with Pex30 probably do not represent LDs. This could be easily verified by co-localization with a blue neutral lipid stain like monodansyl pentane (MDH).

-There is a confusion about what is YFP-MCTP2. In the yeast experiments, only the RHD of MCTP2 is used and the fusion to YFP is termed YFP-MCTP2. In the mammalian cells experiment, it is not clear whether YFP-MCTP2 represents full length MCTP2 or its RHD only. If the expressed MCTP2 is the truncated version, the authors should definitely perform the same experiment with full-length MCTP2. This would clarify if the native MCTP2 shows a similar localization on the ER. If the full-length version has been used instead, then it would be wise to use a different nomenclature (e.g. YFP-MCTP2(RHD) vs. YFP-MCTP2).

In Figure 4 B-c, it is unclear if the peroxisome (CFP-SKL) colocalization on lipid droplets is significant. Although the authors have quantified it and provide a percentage, it is possible that this represents random cooccurrence. Hence, the authors need to show that this number is significant compared to a chance phenomenon. For instance, the authors could generate a random colocalisation measure by "sliding" the CFP-SKL channel by one or 2 microns over the other two channels and remeasure the percentage, to show that it is indeed lower, and that the ~30% colocalization measured is not due to chance.

Minor points

-In Figure 2I, the authors conclude that ER-LD contacts are "altered" based on the movement of Dga1-GFP. They might want to acknowledge the other possibility that the number of LDs might change between the wt and pex30 mutants 24 hours after shift to fresh media.

-Line 69, it should read "Kopito" group and not Kapito

-Generally, the manuscript could describe the experiments a bit more in the text body rather than putting everything in the figure legend. For instance, in Line 135, the authors cite Fig2A-F in one sentence. They give the conclusion of a series of experiments without describing them.

We would like to thank all the reviewers for their thoughtful, constructive comments. Our study has been significantly improved by responding to them.

**Reviewer #1:
Major comments**

One of the most exciting aspects of this manuscript is the identification of a putative Pex30 mammalian ortholog. However, only some preliminary experiments are performed. Other than co-localization of an overexpressed small fragment of YFP-MCTP2 with LiveDrop, there is little characterization of MCTP2 in mammalian cells. Does full length MCTP2 and endogenous MCTP2 localize to these ER subdomains? What happens to lipid droplets and peroxisomes following deletion of MCTP2 or MCTP2 and MCTP1? Is there a similar relationship with seipin? What happens to lipid droplets and peroxisomes in an MCTP2 and seipin double delete? These experiments are essential.

To address these questions, we have done the following.

1. We found that full length MCTP2 fused to GFP colocalizes with LiveDrop in the ER (Fig. 5G). It has a localization that is similar to the fragment of MCTP2 that just contains the reticulon homology domain (RHD) we used in the first version of this study.
2. We knocked down MCTP2 and Seipin by 80% in HeLa cells (Fig. 6A) but we were not able to substantially knock down MCTP1. Knockdown of MCTP2 leads to smaller LDs (Fig. 6B,C), indicating the MCTP2 may have a role in LD biogenesis similar to that of Pex30 in yeast. The effects of knocking down both MCTP2 and Seipin was not different than just knocking down Seipin (Fig. 6B,C). Peroxisomes were not altered significantly by MCTP2 knock down (Supplementary Fig. 5E).
3. We generated a worm strain lacking the only MCTP in this animal. There is a dramatic decrease in LD number and size in the intestines of these animals (Fig. 6D-F).

Together, these results suggest that MCTP2 and Pex30 have similar roles in LD biogenesis. As the reviewer suggests, we would like to visualize endogenous MCTP2 (and MCTP1) but have not yet succeeded. However, we feel the new findings we have been able to add to this study significantly strengthen the case that MCTP2 (and MCTP1 in *C. elegans*) play roles in LD biogenesis that are similar to that of Pex30 in yeast.

In yeast, the lipid droplet membrane and the outer leaflet of the ER are contiguous, allowing exchange of proteins. This has been shown before using FRAP analysis of fluorescently tagged Dga1 and other lipid droplet proteins. The authors propose an interesting possibility that Pex30 may impact ER-LD contacts, but examine this possibility using an unconventional assay involving measurement of the percentage of cells with Dga1-GFP in the ER or in lipid droplets at various growth phases. The localization of Dga1-GFP to these compartments could be affected by many variables and it is not clear that this accurately reflects exchange at ER-lipid droplet contacts. The movement of Dga1-GFP between the compartments should be measured by FRAP in the WT and mutant strains.

As the reviewer suggests, we used FRAP to directly test whether ER-LD connections are altered in cells lacking Pex30 and found they are not (Supplementary Fig. 3I). However, we now show that in cells lacking Pex30 the rate of LD formation significantly decreases (Fig. 3I), which probably explains why the rate of Dga1-GFP relocalization from the ER to LDs is reduced in cells lacking Pex30 (Fig. 3J). These results suggest that Pex30 may alter some property of the ER at LD biogenesis sites, perhaps surface tension, which affects the rate of formation of nascent LDs and LD size.

Are Pex30 and Sei1 (and Nem1) at the same sites in the ER? In the fluorescence microscopy images in Figure 1E/F they seem to be next to each other?

We have performed more careful analysis by quantitatively determining the colocalization of Pex30 with Sei1, and Nem1 using Pearson's coefficient (Fig 2D). Our analysis suggests that there is partial colocalization of Pex30 with Nem1 and Seipin, which are both known to be close to sites of LD biogenesis.

Is it surprising that DAG builds up at sites of lipid droplet biogenesis given that DAG seems to inhibit lipid droplet budding in vitro (M'barek et al. Dev Cell 2017)?

We agree that this is surprising. However, the localization of Nem1 to sites of LD biogenesis suggests that DAG is produced at these sites and could become enriched there when TAG production is induced since DAG is a precursor of TAG. Consistent with this, we (and others) have found that the two enzymes that use DAG to produce TAG (Dga1 and Lro1) become enriched at LD biogenesis sites when LD biogenesis is induced (Choudhary et al, Current Biol, 2018), suggesting that DAG production at these could attract DAG-binding proteins to them.

Some additional speculation about the mechanism by which Sei1 and Pex30 act to promote organelle biogenesis in the conclusion section would be helpful. Why would this specific reticulon be important? Are the defects in biogenesis related to aberrant ER morphology similar to disruption of atlastin?

We have added a brief discussion of how Pex30 and Sei1 might promote organelle biogenesis by altering ER structure and/or lipid metabolism (p. 10-11).

Minor comments

1. "the Kapito group..." should be "the Kopito group..." (line 69)

We have made the correction.

2. "when the cells were grow in..." should be "when the cells were grown in..." (line 124)

We have made the correction.

Reviewer #2:

Major concerns are outlined below:

1. The ostensible reason the authors investigate a role for Pex30 in lipid droplet biogenesis is that they previously showed "10-fold more Pex30 domains than there are PPVs in cells." They "previously identified ER subdomains in *S. cerevisiae* that contain Pex30" and there are "~25 Pex30-containing puncta in the ER per cell." However, this misrepresents scientific record. It was Yan et al., MBoC, in 2008, who showed that Pex30 exists in ER subdomains, and this was further defined by Davids et al., 2013 and Mast et al., 2016 to reveal that there are ~10-fold greater Pex30 domains than peroxisomes, and that these domains can serve as sites for regulating peroxisome biogenesis, both as contact sites for peroxisomes, and also as sites for regulating preperoxisomal vesicle budding from the ER. Only Davids et al., is cited in this manuscript and unfortunately not in a proper context here. This is not the only example. The authors should review the citations carefully to ensure they are appropriate throughout the manuscript.

The reviewer's point is well taken and we now properly cite the papers mentioned at a number of places throughout the manuscript.

2. There are no controls to show that the Erg6-GFP foci in figure 1 correspond to de novo formed lipid droplets.

Jacquier et al., JCS, 2011 showed that Erg6-GFP foci correspond to de novo formed LDs when LD formation is initiated in *GAL1-LRO1* Δ cells. This study is cited.

3. The quantification in figure 1B lacks error bars, and since it is quantifying only those Erg6-GFP foci that have been captured in a single-focal plane through the center of the cell, how reflective is this of the majority of Pex30 foci, which are cortically localized? What would colocalization with other ER markers look like? For example, would Rtn1 which is known to interact with Erg6 have greater or less colocalization than Pex30? What about Sec61?

We have now included error bars in Fig. 1B. The reviewer is concerned that by focusing on the center of the cell we are not getting an adequate view of Pex30 puncta, which are largely in the cortex. We would have liked to take stacks of images of cells expressing endogenously tagged Pex30-2xmCherry so that we could see all the Pex30 puncta but this was not possible we can take only about 3 images before the mCherry bleaches. However, we have taken images focusing on either the center of the cell or the periphery and the extent of colocalization is the same.

4. In figure 1C and throughout this manuscript, Pex30-2xmCherry looks mislocalized to the general tubular ER and not confined to punctate subdomains. How do the authors know that the 2x-mCherry tag isn't altering the dynamics and function of Pex30? This mislocalization clearly alters the interpretation of the colocalization between Pex30 and lipid droplet markers.

Endogenously tagged Pex30-2xmCherry is functional. While *pex30 sei1* mutants have a significant growth defect, Pex30-2xmCherry *sei1* cells grow as well as *sei1* cells (Figure 1 for reviewers). The small fraction of Pex30-2xmCherry that is in the general tubular ER probably reflects the actual localization of Pex30.

5. Given that the authors are relying on a single focal plane from a wide field microscope for most of their experimental work, how are they able to distinguish between objects that truly colocalize and those that are merely adjacent to one another within the ER and between the ER and other organelles? The line tracings shown for Sei1 and Nem1 suggest that these proteins are not colocalized but sit adjacent to each other. How is this unique for Pex30 and not a general feature of all ER membrane proteins?

The reviewers' points are well taken. In the revised version, we are more careful in our use of the term colocalization and instead use association when puncta are near one another but not entirely colocalized. We think the three-color experiments we have added suggest that close associations that are not true colocalizations are still significant. When Pex30-2xmCherry and Sei1-GFP are associated, they were also associated with Erg6-BFP about 95% of the time (Fig. 2 E, F). To address the uniqueness of colocalization, we used Sec13, which forms punctae and marks ER exit sites. Using Pearson's co-efficient we and others (Mast et. al., JBC 2016) have shown that there is little colocalization of Sec13 and Pex30 (Fig. 2B, D), suggesting that the associations of Nem1 and Sei1 with Pex30 are significant.

6. The authors conduct no standard statistical tests (Pearson's, Mander's, etc.) nor do they perform any rigorous object based colocalization analysis, which at a minimum, should be performed for all colocalization analyses in the manuscript.

We have taken reviewers suggestion and performed a careful analysis of colocalization using Pearson's coefficient. We have included quantification in Fig 2D and 5E

7. The authors selectively cite Pagac et al., 2016 to claim that Pex30 physically interacts with Sei1, and yet from the experimental method in that manuscript there can be no claim of specificity, or direct interaction. This claim must be established experimentally as part of this work.

We have removed the citation. Whether Sei1 and Pex30 physically interact is beyond the scope of this study and does not affect our conclusions.

8. The authors have developed a fluorescent "DAG sensor" and yet fail to provide any evidence that it binds DAG, and whether that binding is exclusive to DAG. Until then, these experiments are uninterpretable.

The DAG sensor we are using, the C1a/b domains of PKD, has been well characterized and used in a number of studies; the only way we modified the sensor was to fuse it to a tail anchored transmembrane domain that keeps it in the ER. The sensor has been shown to bind to liposomes with DAG and there is substantial evidence that the affinity of the sensor for membranes in cells reflects changes in DAG concentration (Baron and Malhotra, Science, 2002; Medea et al, EMBO J, 2001; Kim et al, Dev Cell, 2011). We have also characterized the DAG sensor in Choudhary et al, 2018, where we showed that a mutation that ablates binding of the sensor to DAG-containing liposomes prevents the sensor from forming puncta in the ER when LD biogenesis is induced. However, the reviewer is certainly correct that the binding of this sensor (or any lipid sensor) to membranes in cells is complex and binding may be determined by factors in addition to DAG concentration. We have discussed the sensor more carefully in the text.

9. Lipid droplets and peroxisomes are dynamic organelles that respond to changes in cell state and media conditions making comparisons between results from multiple growth conditions difficult to interpret. For example, in figure 2, cells grown to early stationary phase and labeled with BODIPY are compared to cells from mid logarithmic phase and processed for EM. These data should be generated under the same conditions, quantified, and images presented should be representative of the dominant phenotype.

To address these concerns, we stained cells in mid-logarithmic growth phase with BODIPY and checked LD morphology. We have incorporated quantification of the percent of cells with clustered LDs in WT and *pex30* cells in mid-logarithmic and stationary growth phases from three independent experiments (Fig. 3A). We found that there are more clustered LDs in *pex30* cells than in WT in both mid-logarithmic phase and stationary growth phases. In all cases, images are shown are representative of the dominant phenotype.

10. The claim that MCTP2 has a reticulon homology domain similar to Pex30 must be taken at face value as no structural or evolutionary data are presented to verify this claim. Similarly, the authors state that there is no mammalian homologue of Pex30 but do not cite or provide evidence in defense of this claim. These analyses must be presented for critical evaluation.

We now include a description of how we used HHpred to arrive at the conclusion that MCTP2 may contain a reticulon homology domain (p. 11) and more details of the analysis are given in Supplementary Figure 4. We also cite a review that indicates that Pex30 homologues are restricted to fungi (Schluter et al., Mol Biol Evol, 2006).

11. The experimental evidence to support the authors claim that MCTP2 is a functional ortholog of Pex30 is weak. First, the authors show that expressing the RHD domain of MCTP2 tubulates ER membranes in the reticulon deletion strain, as they showed for Pex30 previously, and that expressing this RHD domain in the pex30/sei1 double mutant restores growth. But this doesn't demonstrate that MCTP2 is the functional ortholog of Pex30, it merely demonstrates that the RHD of MCTP2 can tubulate membranes. Does expressing the RHD domain of Pex31, or Rtn1, or any other RHD domain have a similar effect?

We overexpressed Rtn1 and Pex31 in cells that lack Pex30 and Sei1, which rescues the growth defect of this strain (Supplementary Fig. 5A). We have now made appropriate changes to the text stating that MCTP2 is an ER protein that has a reticulon homology domain (RHD).

12. The experiments with the YFP-MCTP2 fusion protein in COS7 cells are difficult to interpret. Overexpression of a tubulating protein will likely have dramatic pleiotropic effects and yet the authors perform no control experiments. In figure 4 and 5 YFP-MCTP2 is punctate, and yet in supplemental figure 3 it is all over the tubular ER. How is the association between this fusion protein and the LiveDrop or PTS reporters unique or functionally relevant? There is too much speculation about the role of this protein.

When cells are transiently transfected with the plasmid expressing YFP-MCTP2(RHD), two localization patterns are seen: at low expression levels the protein is in puncta in the ER (Fig. 5D,E and 7A) while at high levels it is all over the highly curved portions of the ER, like reticulon proteins (Supplementary Fig. 5C). The same is true of Pex30-mCherry in yeast, when expressed at low levels it in puncta but when expressed at high levels it has the same distribution as reticulons, *i.e.*, it is enriched in ER tubules at the edges of ER sheets (Joshi et al, 2016). With regard to function, we now show that depletion of MCTP2 in mammalian cells and knocking out MCTP in *C. elegans* alters LD size and abundance, which suggests that MCTP2 plays roles in LD formation and lipid metabolism that are similar to those of Pex30 in yeast. We have reduced the amount of speculation in the text.

13. The example in yeast presented in figure 5a, is unconvincing. There is no quantification and even if there were what conclusion could be drawn? These cells have no peroxisomes and are deficient in autophagy. The Pex14 foci may be a PPV or it may an attempt by the cell to sequester PMPs away from other membranes to prevent unwanted affects. Is the ER normal in these cells? Is the UPR upregulated?

The van der Klei group demonstrated that the Pex14 foci are PPVs that mature into functional peroxisomes when Pex3 is re-expressed in cells lacking Pex3 and Atg1 (Knoops et al, JCB, 2014). The ER is normal in cells lacking these cells and Pex14 puncta not on the ER are in small vesicles that are probably PPVs (Josi et al, JCB,

2016). We do not know whether the UPR is upregulated but we do not see ER expansion, which often accompanies UPR induction (Schuck et al, JCB, 2009). To address the reviewer's concerns, we now quantitate the percent of cells that exhibit colocalization between all three punctae (Pex30-2xmCherry, Pex14-YFP (PPVs), the LD marker Erg6-BFP) in WT and *pex3 atg1* cells (Fig. 7B, C).

Reviewer #3:

Major points

-The claim that Pex30 recognizes DAG-enriched domains is not properly supported. In Figure 1H, the authors present a biosensor for ER DAG. Its robustness should be validated further. Could it bind TAGs as well? It is important to validate that the signal obtained is specific, for example, by including a C2 domain mutant.

Additionally, the lipid strip experiment in Supp Fig 1D, hardly supports the claim that Pex30 could be regulated by DAG accumulation. Pex30 seems to bind PC or the "control" (whatever it is) better. The authors should clarify this and specify what the "control" is. Are there any other non-labelled lipid species on the strip?

Altogether, I think that all mention of DAG should be removed unless better validation is provided for both the biosensor and the lipid strip experiment.

The DAG sensor has been shown to bind to liposomes with DAG and there is substantial evidence that the affinity of the sensor for membranes in cells reflects changes in DAG concentration (Baron and Malhotra, Science, 2002; Medea et al, EMBO J, 2001; Kim et al, Dev Cell, 2011). We have also characterized the DAG sensor in Choudhary et al, 2018, where we showed that a mutation that ablates binding by the sensor to DAG-containing liposomes prevents the sensor from forming puncta in the ER when LD biogenesis is induced. We have not investigated whether the sensor binds to TAG in addition to DAG, a possibility we mention in the text (p. 7). In the previous version of this manuscript, we did not mean to suggest that Pex30 localization in the ER is affected by DAG distribution, only that Pex30 might bind DAG. We have removed the lipid strip data from the manuscript. Whether Pex30 binds lipids and, if so, which domains are necessary are interesting questions but a topic for future efforts.

-In Figure 1E, the authors conclude that Sei1 puncta not co-localizing with Pex30 probably do not represent LDs. This could be easily verified by co-localization with a blue neutral lipid stain like monodansyl pentane (MDH).

We addressed this concern by examining the localization of Sei1-GFP, Pex30-2xmCherry, and the LD marker BFP-Erg6 (Fig. 2E,F). This shows that Sei1 puncta associated with Pex30 are also associated with Erg6-BFP (LDs).

-There is a confusion about what is YFP-MCTP2. In the yeast experiments, only the RHD of MCTP2 is used and the fusion to YFP is termed YFP-MCTP2. In the mammalian cells experiment, it is not clear whether YFP-MCTP2

represents full length MCTP2 or its RHD only. If the expressed MCTP2 is the truncated version, the authors should definitely perform the same experiment with full-length MCTP2. This would clarify if the native MCTP2 shows a similar localization on the ER. If the full-length version has been used instead, then it would be wise to use a different nomenclature (e.g. YFP-MCTP2(RHD) vs. YFP-MCTP2).

We apologize that the previous version of the manuscript was unclear. We now show localization of both the truncated version of MCTP2 with only the RHD, called YFP-MCTP2(RHD), and full-length MCTP2 (GFP-MCTP2). Both fusions have similar localizations and are enriched in puncta that frequently co-localize in the ER with LiveDrop (Fig. 5E-G).

In Figure 4 B-c, it is unclear if the peroxisome (CFP-SKL) colocalization on lipid droplets is significant. Although the authors have quantified it and provide a percentage, it is possible that this represents random co-occurrence. Hence, the authors need to show that this number is significant compared to a chance phenomenon. For instance, the authors could generate a random colocalization measure by "sliding" the CFP-SKL channel by one or 2 microns over the other two channels and remeasure the percentage, to show that it is indeed lower, and that the ~30% colocalization measured is not due to chance.

To address this concern, we rotated the CFP-SKL by 90 degrees clockwise and quantified the percent association. Since there is little colocalization after rotation, suggesting that the 30% colocalization is not due to chance.

Minor points

-In Figure 2I, the authors conclude that ER-LD contacts are "altered" based on the movement of Dga1-GFP. They might want to acknowledge the other possibility that the number of LDs might change between the wt and pex30 mutants 24 hours after shift to fresh media.

The reviewer makes a good point. We confirmed that nascent LDs are indeed generated in cells lacking Pex30 (Fig. 3I, J)

-Line 69, it should read "Kopito" group and not Kapito

We have made the correction.

-Generally, the manuscript could describe the experiments a bit more in the text body rather than putting everything in the figure legend. For instance, in Line 135, the authors cite Fig2A-F in one sentence. They give the conclusion of a series of experiments without describing them.

The revised manuscript contains more substantial descriptions of the experiments in the Results section.

Figure 1 for Reviewers: Pex30-2xmCherry is functional: Strains were grown to mid-logarithmic growth phase, serially diluted, spotted on to the YPD plates and incubated at 23°C for 3 days.

REVIEWERS' COMMENTS:

Reviewer #1 (Remarks to the Author):

My concerns have been sufficiently addressed. The new data indicating that full length MCTP localizes to live droplet-positive puncta and that loss of MCTP results in defects in lipid droplets in cultured cells / *C. elegans* are consistent with MCTP acting as the functional ortholog in human cells. Although the current analysis of MCTP1/2 leaves many questions unanswered, these seem better addressed in future papers. Overall, the manuscript is interesting and makes an important contribution to the field. I recommend acceptance and publication.

Reviewer #2 (Remarks to the Author):

In this revised and expanded manuscript, Joshi and colleagues present evidence for a role for Pex30 in the biogenesis of lipid droplets. Previously, they showed that Pex30 has a reticulon-homology domain that is capable of tubulating ER membranes. Pex30 demarcates sites of peroxisome biogenesis in the ER. The authors extend this function of Pex30 to lipid droplets by showing that: Pex30 colocalizes with sites of de novo lipid droplet biogenesis; Pex30 has negative genetic interactions with Sei1, a protein involved in lipid droplet biogenesis; the reticulon homology domain of Pex30 can rescue the synthetic lethal interaction; lipid droplet biogenesis is delayed, and lipid droplets have altered morphology in Pex30 deletion mutants. They also identify a metazoan protein MCTP2 as having a reticulon homology domain similar to Pex30 and demonstrate that it functions in an analogous manner in regulating lipid droplets in Cos7 cells and *C. elegans*.

The advancement of the idea that the ER is partitioned by distinct classes of reticulon homology domain containing proteins is an important contribution to our understanding of how the ER is spatially regulated, and how peroxisome biogenesis and lipid droplet biogenesis may be coordinated. The authors have addressed most of my previous concerns and I recommend acceptance of this manuscript after consideration of the points below with appropriate revision to the text.

1. The interpretation of the authors experiments with the *pex3atg1* deletion strain should be critically reexamined. As established by the van der Klei group, the idea that peroxins and peroxisomal membrane proteins could be packaged into vesicles in the absence of Pex3 was an important advance for the field. However, as I previously mentioned, it is difficult to interpret what the Pex14 foci are in these cells. The cells are deficient in autophagy and peroxisome biogenesis. From the fluorescence microscopy images, it is not possible to distinguish between PPVs and foci on the ER, as demonstrated by the overlap of ~ half of the Pex14 foci with Pex30. Subcellular fractionation or immunoEM would be required to delineate between these possibilities. Including these caveats in the text would be beneficial.

2. On page 4, 15th line, citation is lacking for the function of Pex30 in regulating peroxisome size and number.

3. Ensure all acronyms are properly introduced. On page 5, SE and TAG are presented without first delineating what the acronyms mean. This will aid comprehension for the non-specialist and I would double check that all other acronyms are properly introduced.

4. On page 5, first line, it should read "the multiple C2 domain containing transmembrane protein".

5. The expanded description of the DAG sensor, including the authors recent publication showing that a point mutation in the C1a/b domain abolishes binding, is appreciated. However, it is not clear whether the domain is recognizing DAG (or TAG) in cis or trans. How do the authors know that this domain is specifically recognizing DAG in the ER?
6. The authors mention in their rebuttal that they have taken images of the periphery of cells, but do not show these in the majority of their figures. Figure 1, in particular, would benefit from their inclusion, also Figure 2A, B, and C.
7. In Figures 2 A and C, the line tracing should be extended to show the profile for the left colocalization. It appears that the fluorescence signals are not-overlapping but adjacent to each other.
8. Error bars are missing in figure 2 F.
9. In figure 2 G, Erg6 looks dramatically different from the images of Erg6 in figure 1. Are there more consistent, appropriate, images the authors could use instead?
10. The legend of figure 5 should be rewritten. Given that the function of Pex30 is unknown, i.e., what the tubulation of ER membranes by Pex30 is contributing to the biogenesis of peroxisomes and lipid droplets. The data are too preliminary at this time to draw the conclusion that MCTP2 is the functional homologue of Pex30. Also, despite having a similar RHD the topology of the proteins is quite different.
11. The sentence starting the paragraph on page 14, line 7, should be corrected. Do the authors intend to say, "complete deletion"?
12. The data for a role for MCTP2 in lipid droplet biogenesis are more compelling than a role for MCTP2 in peroxisome biogenesis. The association shown in Figure 7 may be non-random, but consequences from lack of MCTP2 on peroxisome distribution, numbers, etc. are not apparent from its knockdown.

Reviewer #3 (Remarks to the Author):

The authors have addressed and/or clarified all of the points raised adequately. This is a very interesting story.

Response to reviewer#2:

1. The interpretation of the authors experiments with the pex3atg1 deletion strain should be critically reexamined. As established by the van der Klei group, the idea that peroxins and peroxisomal membrane proteins could be packaged into vesicles in the absence of Pex3 was an important advance for the field. However, as I previously mentioned, it is difficult to interpret what the Pex14 foci are in these cells. The cells are deficient in autophagy and peroxisome biogenesis. From the fluorescence microscopy images, it is not possible to distinguish between PPVs and foci on the ER, as demonstrated by the overlap of ~ half of the Pex14 foci with Pex30. Subcellular fractionation or immunoEM would be required to delineate between these possibilities. Including these caveats in the text would be beneficial.

Point taken. We now include a statement, "These Pex14-GFP puncta could either be present on the ER membrane or PPVs."

2. On page 4, 15th line, citation is lacking for the function of Pex30 in regulating peroxisome size and number.

We have now included the appropriate citation.

3. Ensure all acronyms are properly introduced. On page 5, SE and TAG are presented without first delineating what the acronyms mean. This will aid comprehension for the non-specialist and I would double check that all other acronyms are properly introduced.

We defined SE and TAG in the introduction on page 3 and do not think it is necessary to define them again in the results section on page 5 unless the editors disagree.

4. On page 5, first line, it should read "the multiple C2 domain containing transmembrane protein".

Done.

5. The expanded description of the DAG sensor, including the authors recent publication showing that a point mutation in the C1a/b domain abolishes binding, is appreciated. However, it is not clear whether the domain is recognizing DAG (or TAG) in cis or trans. How do the authors know that this domain is specifically recognizing DAG in the ER?

We cannot rule out that the sensor binds DAG in trans (i.e., binding to a membrane near the ER). However, the linker between the C1a/b domain is only a few amino acids making in trans binding unlikely. If the ER DAG-sensor were to bind in trans, the most likely organelles it would bind would be LDs or vacuoles (since the cytoplasmic sensor mostly binds vacuoles). There is no enrichment of the ER-DAG sensor on LDs or vacuoles in wild-type cells, suggesting that it does not normally bind in trans. It is possible but not likely that the sensor binds in trans in cells lacking pex30.

6. The authors mention in their rebuttal that they have taken images of the periphery of cells, but do not show these in the majority of their figures. Figure 1, in particular, would benefit from their inclusion, also Figure 2A, B, and C.

We have included the images of the periphery of cells in supplementary figure 1C.

7. In Figures 2 A and C, the line tracing should be extended to show the profile for the left colocalization. It appears that the fluorescence signals are not-overlapping but adjacent to each other.

We have extended the line tracing and included the plot profile in Figure 2A and 2C.

8. Error bars are missing in figure 2 F.

We have included the error bars in Figure 2F.

9. In figure 2 G, Erg6 looks dramatically different from the images of Erg6 in figure 1. Are there more consistent, appropriate, images the authors could use instead?

In Figure 1, the new LD formation is induced in cells devoid of LDs. In the absence of LDs, Erg6-GFP is mainly localized on the ER (upper panel, Figure 1A) however, Erg6-GFP localizes to LD forming sites upon induction (lower panel, Figure 1A). Cells used in in figure 2G are wild-type that have LDs. Therefore, the Erg6-BFP is localized to LDs and not in the ER. Hence, the images look different in Figure 2G and Figure 1A.

10. The legend of figure 5 should be rewritten. Given that the function of Pex30 is unknown, i.e., what the tubulation of ER membranes by Pex30 is contributing to the biogenesis of peroxisomes and lipid droplets. The data are too preliminary at this time to draw the conclusion that MCTP2 is the functional homologue of Pex30. Also, despite having a similar RHD the topology of the proteins is quite different.

We disagree. We show that the reticulon homology domain (RHD) of MCTP2 can functionally replace Pex30 in yeast, has the same distribution in the ER as Pex30, and mammalian cells depleted of MCTP2 (or worms lacking MCTP) have altered LD number and size, just like yeast cells lacking Pex30. We think these findings are strong evidence they MCTP2 and Pex30 have similar functions. In addition, the proteins almost certainly have the same topology; RHDs bind the cytoplasmic leaflet of membranes and face the cytoplasm. There is no reason to think that any part of Pex30 or MCTP2 faces the ER lumen. However, because we do not have definitive proof that MCTP2 and Pex30 have the same functions, we have altered the title of Figure 5.

11. The sentence starting the paragraph on page 14, line 7, should be corrected. Do the authors intend to say, "complete deletion"?

We have corrected the typographical error.

12. The data for a role for MCTP2 in lipid droplet biogenesis are more compelling than a role for MCTP2 in peroxisome biogenesis. The association shown in Figure 7 may be non-random, but consequences from lack of MCTP2 on peroxisome distribution, numbers, etc. are not apparent from its knockdown.

The reviewer is correct. From the knockdown of MCTP2, the distribution and numbers of peroxisomes are not affected. However, it might affect the size of peroxisomes or the association of peroxisomes to LDs and the ER subdomains, which will be addressed in the future.